# Research

ecology, evolution, physiology

cellular respiration, climate change, developmental plasticity, heat-stress, Oroboros, thermal acclimation

**Author for correspondence:**
Mylene M. Mariette
e-mail: m.mariette@deakin.edu.au,
mylene.mariette@ebd.csic.es

<sup></sup>†Present address: School of Research Biology, The Australian National University, Canberra, ACT 2601, Australia

# Prenatal acoustic programming of mitochondrial function for high temperatures in an arid-adapted bird

Eve Udino[1], Julia M. George[2], Matthew McKenzie[1], Anaïs Pessato[1], Ondi L. Crino[1,†], Katherine L. Buchanan[1] and Mylene M. Mariette[1,3]

[1]School of Life and Environmental Sciences, Deakin University, Waurn Ponds, Victoria 3288, Australia
[2]Department of Biological Sciences, Clemson University, Clemson, SC 29634, USA
[3]Estación Biológica de Doñana EBD-CSIC, Seville, 41092, Spain

EU, 0000-0002-9794-2795; JMG, 0000-0001-6194-6914; MM, 0000-0001-7508-1800; AP, 0000-0002-9694-2013; OLC, 0000-0001-5700-1387; KLB, 0000-0002-6648-5819; MMM, 0000-0003-0567-4111

Sound is an essential source of information in many taxa and can notably be used by embryos to programme their phenotypes for postnatal environments. While underlying mechanisms are mostly unknown, there is growing evidence for the involvement of mitochondria—main source of cellular energy (i.e. ATP)—in developmental programming processes. Here, we tested whether prenatal sound programmes mitochondrial metabolism. In the arid-adapted zebra finch, prenatal exposure to 'heat-calls'—produced by parents incubating at high temperatures—adaptively alters nestling growth in the heat. We measured red blood cell mitochondrial function, in nestlings exposed prenatally to heat- or control-calls, and reared in contrasting thermal environments. Exposure to high temperatures always reduced mitochondrial ATP production efficiency. However, as expected to reduce heat production, prenatal exposure to heat-calls improved mitochondrial efficiency under mild heat conditions. In addition, when exposed to an acute heat-challenge, *LEAK* respiration was higher in heat-call nestlings, and mitochondrial efficiency low across temperatures. Consistent with its role in reducing oxidative damage, *LEAK* under extreme heat was also higher in fast growing nestlings. Our study therefore provides the first demonstration of mitochondrial acoustic sensitivity, and brings us closer to understanding the underpinning of acoustic developmental programming and avian strategies for heat adaptation.

## 1. Introduction

Developmental programming occurs across taxa, as early life environments alter developmental processes and thereby shape individual phenotype [1,2]. Beyond the negative consequences of poor early life conditions [3], programming can improve individual fitness by tailoring phenotypes to future environments, predicted from anticipatory cues [4,5]. Intriguingly, prenatal sound is emerging as an alternative source of information for embryos, in addition to the well-known effects of maternal (e.g. hormones and nutrition) or environmental (e.g. predator scent) biochemical cues. From invertebrates to birds, embryos use external acoustic cues to adaptively alter their developmental trajectories [6]. Prenatal sound and vibrations can shift hatching time in all oviparous taxa, but also directly impact individual cognition and physiology [6]. This may occur through neurological or epigenetic changes [6], or even through direct alteration of cell physiology in plants [7,8]. Whether such cellular responses also occur in animals is unknown, but could potentially arise indirectly through changes in mitochondrial function.

Mitochondria provide most of the energy used by organisms (in the form of adenosine trisphosphate, ATP) and concomitantly generate endogenous heat [9]. Described as the 'power-house of the cell', their role in evolutionary and ecological processes is increasingly recognized [10], including for developmental programming [11]. As early life conditions affect mitochondrial function on the long-term, and across tissues, mitochondrial alteration has been suggested as one overarching mechanism underlying the orchestrated programming of multiple functions and traits in developing organisms [11]. This hypothesis, recently formulated for the origins of human diseases [11], probably extends to adaptive variation in wildlife. Indeed, mitochondrial function is increasingly understood as underlying inter-individual variation in life-history and secondary sexual traits, as well as in performance and fitness [12–14]. Whether this variation arises from developmental programming of mitochondrial function remains to be tested. Furthermore, whether prenatal sound could programme mitochondria is unknown. Such acoustic mitochondrial programming may notably occur through the effects of sound on stress physiology [6], which in turn is known to impact mitochondrial function during development [11,15]. In addition, acoustic sensitivity of mitochondria is suggested in the yellow-legged gull (*Larus michahellis*), where embryonic exposure to alarm calls affected hatchling physiology, as well as mitochondrial DNA copy number, a proxy for mitochondrial content [16]. It is therefore possible that some of the adaptive effects of prenatal sound across taxa [6] are driven or accompanied by alteration of mitochondrial function.

Mitochondrial function notably varies adaptively in response to environmental temperatures, and has played an essential role in the evolution of endothermy in vertebrates [17,18]. During development, mitochondrial respiration rates increase with incubation temperature in reptile and poultry embryos [19,20] and with postnatal nest temperature in songbirds [21]. While the potential adaptive benefits of these developmental changes are unknown, adult mitochondrial function varies adaptively with thermal acclimation, as does whole-organism metabolism [22–26]. Indeed, mitochondrial respiration is thought to determine organismal thermal limits [27], and thus potentially, their ability to adapt to increasing temperatures under climate change. Nonetheless, unlike at low temperatures (e.g. [23,25,28]), mitochondrial responses to heat have received limited attention and appear inconsistent, under both acute [29–31] and chronic [32,33] exposure.

Multiple aspects of mitochondrial function can respond to environmental temperatures in both mature and developing organisms. Mitochondria use oxygen to oxidize food-derived substrates and thereby transfer protons into the mitochondrial inter-membrane space; these protons then re-enter the mitochondrial matrix through the ATP-synthase, which produces ATP. The overall rate of mitochondrial respiration (measured through enzymatic activity or directly as $O_2$ consumption) and/or mitochondrial density can vary plastically with temperature [22,25], including during development [19,21]. Furthermore, and importantly for adaptive responses, a proportion of respiration, diverted from ATP production towards non-phosphorylating 'LEAK' respiration, can also vary [34]. LEAK respiration occurs when protons bypass the ATP-synthase and instead leak back into the mitochondrial matrix passively or through uncoupling proteins [10]. Even though it reduces the efficiency of ATP production,

LEAK is an essential source of heat production, in mammals, but also potentially in birds [17,34]. LEAK can also reduce the oxidative damage associated with the production of reactive oxygen species (ROS) in mitochondria ('uncoupling to survive hypothesis' [35]). This leads to trade-offs between energy production, heat generation and oxidative balance [36], with LEAK accounting for 20–70% of cellular respiration, or up to 25% of the organismal basal metabolic rate [28,34,37]. LEAK proportion and therefore mitochondrial efficiency (inverse to the ratio of LEAK over mitochondrial overall respiration under endogenous (ROUTINE) or stimulated (ETS) conditions) can differ between species [12], but also between individuals [13]. Notably, mitochondrial efficiency may underlie variation in individual quality, including growth performance, since high efficiency reduces the amount of food needed to sustain growth [13]. Alternatively, variations in mitochondrial efficiency may correspond to differential adaptive strategies [10,12], with, for example, thermal acclimation to cold in fish, mammals and birds [28,38,39] increasing the proportion of LEAK respiration. During development, the proportion of LEAK also varies, notably with stress exposure decreasing mitochondrial efficiency [11,15], although no anticipatory adaptive programming has so far been investigated.

Here, we tested whether adaptive programming of mitochondrial function for high temperatures occurs, through prenatal acoustic signals. In the zebra finch (*Taeniopygia guttata*), parents emit 'heat-calls' during incubation at high temperatures [40,41], which improves the caller's thermoregulation capacity [42]. Embryonic exposure to such heat-calls adaptively alters nestling development, with nestlings being smaller in hot nests, but larger in milder shady conditions, which then improves reproductive success throughout adulthood [40]. Prenatal heat-call exposure also shifted individual thermal preference towards hotter breeding nests as adults [40]. As a potential mechanistic pathway for these effects, we hypothesized that prenatal exposure to heat-calls programmes mitochondrial function, to suit the forecasted hot conditions post-hatch. We exposed embryos to playbacks of either heat-calls or control-calls, and manipulated foster rearing nest temperatures. At 13 days post-hatch, we measured nestling mitochondrial function in intact red blood cells (RBCs) in two distinct experiments, with nestlings under either: (i) undisturbed nest conditions, or (ii) acute heat-challenge (i.e. 2.5 h in a heated-chamber, stepped up to 44°C). We predicted that (i) if prenatal programming of mitochondrial function by heat-calls is aimed at reducing heat production, mitochondrial efficiency would be higher in heat-call exposed birds. Indeed, in addition to reducing leak-related heat production, high mitochondrial efficiency, by lowering food requirements [13], would decrease the amount of heat generated by food digestion (heat-increment of feeding [43,44]). Alternatively, or in addition, (ii) if the adaptive benefits of heat-call exposure stem from reducing the detrimental physiological impact of heat, heat-call individuals may increase LEAK at very high temperatures to decrease oxidative damage. Furthermore, we predicted that (iii) if high temperatures, in the nest or the heated-chamber, represent a stressor for the individual, this would decrease mitochondrial efficiency, either because of mitochondrial function impairment or because LEAK increases to reduce the oxidative impact of stress. Lastly, to determine if our observations matched the adaptive mitochondrial

responses hypothesized above, we investigated how nestling growth rate relates to mitochondrial efficiency and other mitochondrial traits.

## 2. Material and methods

### (a) Prenatal acoustic playback

The study was carried out in summer from December 2018 to April 2019 at Deakin University, Geelong, Australia. Fifty-nine male and 52 female wild-derived zebra finches were allowed to breed freely in a large outdoor aviary ($12 \times 6 \times 3$ m). We followed the same breeding procedure as described in Mariette & Buchanan [40] on a different cohort of birds. Briefly, eggs were collected on the laying day and incubated in a fan-less artificial incubator (Bellsouth 100 electronic incubator) at 37.5°C and 60% humidity. On the 10th day of incubation, eggs from the same clutch were randomly transferred to one of two experimental incubators and exposed daily to either heat-calls (treatment) or contact calls (control) until hatching (electronic supplementary material, figure S1). To ensure normal stimulation of the auditory system, both groups were also exposed to whine calls, another parental contact call with a complex acoustic structure. The playbacks were broadcast between 10.00 and 18.00 at 65 dB from two speakers (Sennheiser HD439) inside the incubator, and externally connected to an amplifier (Digitech 18 W) and an audio player (Zoom H4nSP and Marantz PMD670). Egg trays and audio players were swapped daily between the two playback incubators, to avoid any incubator-specific effects. At hatching, nestlings were given to foster (non-genetic) parents for rearing. To minimize differences in postnatal parental provisioning, we homogenized all brood sizes to three to four nestlings, with individuals from both prenatal playback groups in each nest.

### (b) Postnatal nest temperature

The daytime temperature inside the nest-boxes (10.00–18.00) was recorded and manipulated throughout nestling development (hatching to 13 day post-hatch, dph) to create a gradient of cool to hot nests (details in the electronic supplementary material). The mean nest temperature, hereafter '12D-$T_{nest}$', was then calculated from hatching to 12 dph for each individual in a brood, during the warmest part of the day (11.00–18.00). In addition, we extracted the short-term nest temperature before blood sampling, as the temperature: overnight (average from 00.00 to 7.00), at sunrise (i.e. minimal night temperature, occurring between 6.30 and 7.30), or 3 h before sampling ('morning nest temperature', AM-$T_{nest}$, occurring between 8.30 and 9.45).

### (c) Experimental conditions for in-nest and heat-challenge experiments

All the nestlings were weighed between 15.00 and 15.30 at 7 and 12 dph to determine their growth rate, measured as mass gain between these two time points. At 13 dph, nestlings were randomly allocated to either the 'in-nest' or 'heat-challenge' experiment, making sure there was no overall bias in nestling mass or hatching order. Each day, we sampled one nestling for each experiment from the same playback group, from either the same or different rearing nests, and balanced sample sizes for nest temperature across days of the experiment.

In-nest birds were taken from their nest between 11.30 and 12.45 (average 12.12), just prior to blood sampling. In the heat-challenge experiment, birds were exposed to a standardized stepped profile of temperature increase, in a heated-chamber over a 2.5 h period (as part of a separate experiment). They were taken from the nest at 9.30 and food deprived until blood sampling. After 30 min at room temperature (25°C) and 35 min of acclimation in the heated-chamber at 29°C, they were exposed to 20–30 min stages at 35, 40, 42 and 44°C, before temperature was dropped back to 35°C for the final 15 min of exposure. Two individuals showing signs of severe heat-stress at 42°C were not exposed to 44°C.

Within 3 min following the end of the heat-challenge or collection from the nest, nestlings were euthanized using isoflurane and decapitation to collect tissues for a separate study. Blood was collected from the jugular vein and kept on ice until processing within 30 min.

### (d) Mitochondrial function measurements in intact RBCs

Mitochondrial function in fresh intact RBCs was measured in 46 samples using an Oxygraph-2 k high-resolution respirometer (Oroboros Instruments, Innsbruck, Austria), following standard procedures [15,37,45,46]. Briefly, the RBC resuspensions were prepared, then added to one of the respirometer chambers pre-equilibrated at 40°C to successively measure several mitochondrial respiration rates (details in electronic supplementary material, figure S2). We first measured (i) basal respiration for 10 min ('ROUTINE', mitochondrial $O_2$ consumption under endogenous conditions); then, (ii) 'LEAK' respiration ($O_2$ consumption associated with non-phosphorylating respiration) by adding oligomycin (5 mM; Sigma O4876), an inhibitor of the ATP-synthase; (iii) the maximal working capacity of the electron transport system ('ETS', maximal stimulated $O_2$ consumption) by adding FCCP (carbonyl cyanide-p-tri-fluoro-methoxyphenyl-hydrazone, 50 µM, titration, Sigma C2920), a mitochondrial uncoupler and (iv) non-mitochondrial respiration by adding antimycin A (0.7 mM; Sigma A8674).

For rate calculations, non-mitochondrial respiration was subtracted from all respiration rates. Then, the mitochondrial respiration associated with ATP production (oxidative phosphorylation; OXPHOS) was obtained by subtracting LEAK from ROUTINE. The respiration rates were normalized with the protein content (TP) of the samples (details in the electronic supplementary material). Three flux control ratios (FCRs) were calculated from these mitochondrial rates. $FCR_{L/R}$ (LEAK/ROUTINE) corresponds to the proportion of LEAK relative to total mitochondrial respiration under basal conditions (i.e. lower values indicate higher efficiency to produce ATP and lower endogenous heat production). $FCR_{L/ETS}$ (LEAK/ETS) represents the fraction of proton leak relative to the ETS maximal capacity. Lastly, $FCR_{R/ETS}$ (ROUTINE/ETS) represents the mitochondrial respiratory capacity, or how close endogenous mitochondrial respiration is to the maximal respiratory capacity (i.e. a 'working pace').

### (e) Statistical analyses

From the 46 individuals sampled, we successfully obtained mitochondrial rates for 39 individuals. Two samples had to be excluded for insufficient protein content; one was assayed at the wrong temperature; and four had ROUTINE ≫ ETS (non-biological values). In this final sample size, there were 20 and 19 birds in the in-nest and heat-challenge experiments, respectively.

To test whether $O_2$ consumption varied between the different respiration rates, we ran linear mixed models (LMMs) with $O_2$ consumption rate as a response variable, the type of respiration rate (i.e. ROUTINE, OXPHOS, LEAK or ETS) as a categorical fixed effect and bird identity as a random effect. Post hoc Tukey tests were conducted using the glht function from multcomp package.

To test the effects of developmental conditions (prenatal playback and postnatal acclimation temperature 12D-$T_{nest}$), under undisturbed in-nest conditions at sampling (i.e. in-nest experiment), we ran separate LMM including each mitochondrial parameter as a response variable, and the prenatal playback, 12D-$T_{nest}$ and their interaction as main effects, with morning nest temperature (AM-$T_{nest}$) as a covariate. We used AM-$T_{nest}$ in all models rather than overnight or sunrise temperature because AM-$T_{nest}$ had a better model fit overall (i.e. lower Akaike's information criterion corrected for small sample sizes, AICc). We

included brood identity as a random effect, unless its variance was null. Since AM-$T_{nest}$ had stronger effects than 12D-$T_{nest}$ (see results), as a confirmatory analysis, we re-ran models to test the interaction between playback and AM-$T_{nest}$ (instead of with 12D-$T_{nest}$), which was also always non-significant.

For the heat-challenge experiment, we ran exactly equivalent LMMs, but considering as predictors the heated-chamber temperature deviations from the mean nest temperature (i.e. chamber temperature above 12D-$T_{nest}$; $\Delta$12D-$T_{nest}$) and the morning nest temperature (chamber temperature above AM-$T_{nest}$; $\Delta$AM-$T_{nest}$). In both experiments, including clutch as random factor (to account for genetic similarities) instead of brood, did not affect results (electronic supplementary material, table S1).

For both experiments, to test how mitochondrial parameters relate to nestling growth while accounting for brood size, we ran separate LMMs including growth rate as the response variable and one of the mitochondrial parameters as a fixed effect, together with brood size, prenatal playback, 12D-$T_{nest}$ and the playback by 12D-$T_{nest}$ interaction, and including brood identity as a random factor. Even though we standardized brood sizes to three or four nestlings, three individuals (i.e. 7.7% of sample size) were in a brood size of two nestlings, due to early mortality; we thus ran the models with and without these three birds. Then, to test whether the observed effects (see Results) were better explained by final body size rather than actual growth rate, we ran the same models, but using nestling mass at 12 dph as the response.

All models were fitted in R (v. 4.0.1), using the *lme4* and *lmerTest* packages. Continuous predictors were scaled, and normality and homoscedasticity of residuals visually inspected. Collinearity among predictors was tested in the full models by calculating the variance inflation factors (VIFs) [47]. We adopted a backward stepwise procedure to remove non-significant terms, starting with interactions, to obtain the model with the lowest AICc. Full models and AICcs comparisons with reduced models are presented in electronic supplementary material, tables S2–S6.

## 3. Results

### (a) In-nest conditions

Under undisturbed nest conditions, consistent with the hypothesis that heat-calls reduce heat production, *OXPHOS* (producing ATP) was significantly higher and mitochondrial efficiency, measured as either $FCR_{L/R}$ or $FCR_{L/ETS}$, significantly higher in heat-call than in control-call birds (table 1 and figure 2). *LEAK* respiration rate was not directly affected, and was significantly higher than *OXPHOS* in zebra finch nestlings (figure 1; pairwise comparison: $p = 0.003$). Higher efficiency in heat-call birds was observed across nest temperatures (i.e. no interaction of playback with either 12D-$T_{nest}$ or AM-$T_{nest}$; electronic supplementary material, table S2).

However, as predicted, high nest temperatures negatively affected mitochondrial efficiency, particularly on the short-term. The average daytime temperature experienced from hatching to 12 dph (12D-$T_{nest}$) only had mild influence, with mitochondrial efficiency tending to be lower in hotter nests (i.e. higher $FCR_{L/ETS}$; table 1 and figure 2b). More notably, the morning temperature before testing (AM-$T_{nest}$) negatively affected *OXPHOS* (and *ROUTINE* marginally), leading to significantly lower efficiency (higher $FCR_{L/R}$) on hotter mornings (table 1; electronic supplementary material, figure S3A–C).

Mitochondrial efficiency, and other mitochondrial parameters, were unrelated to nestling growth rate (i.e. mass gain between D7 and D12) or final mass (at day 12; electronic supplementary material, tables S7 and S8). While unexpected,

this is consistent with mitochondrial function changing with short-term (i.e. morning) temperatures, and thermal conditions at sampling (before the heat of the day) only representing a fraction of conditions experienced throughout the day.

### (b) Acute heat-challenge

When nestlings were exposed to a standard acute heat-challenge just prior to cellular respiration measurement, the heated-chamber conditions represented a more extreme deviation from in-nest conditions (i.e. comparatively hotter chamber; $\Delta$12D-$T_{nest}$) for individuals from cool than hot nests. As observed in the in-nest experiment, overall, mitochondrial efficiency significantly decreased (higher $FCR_{L/ETS}$) under hotter chamber conditions (relative to 12D-$T_{nest}$ or AM-$T_{nest}$; figure 3; electronic supplementary material, figure S3), and *LEAK* increased in more extreme heat (relative to 12D-$T_{nest}$; table 1 and figure 3). This pattern, however, differed between playback groups, as there was a significant interaction between playback and $\Delta$12D-$T_{nest}$ on mitochondrial efficiency, for both $FCR_{L/R}$ and $FCR_{L/ETS}$ (table 1 and figure 3). For control-call birds, mitochondrial efficiency again decreased in relatively hotter chambers, whereas heat-call birds were more stable across the gradient of temperature differential (slight decrease), but did not have higher efficiency than controls overall. In fact, *LEAK* tended to be higher in heat-call birds ($p = 0.055$; table 1).

Interestingly however, *LEAK* ($t = 2.84$, $p = 0.014$), *ROUTINE* ($t = 2.45$, $p = 0.043$) and to a lesser extent *ETS* ($t = 2.34$, $p = 0.054$) were all higher in faster growing nestlings, while *OXPHOS* was not (figure 4; electronic supplementary material, table S9). This would indicate that increasing *LEAK* under extreme heat was not a sign of nestlings in poor condition, but may instead allow reduction of oxidative damage in individuals where ROS levels are expected to be higher (i.e. fast-growing individuals [48]). Accordingly, no such correlation between *LEAK* (or other mitochondrial traits) and nestling final mass was found (electronic supplementary material, table S10), confirming that effects were not driven by larger or more developed individuals (at 12 days old) having more sustained mitochondrial activity.

## 4. Discussion

We found that an acoustic signal experienced prenatally shaped individual mitochondrial function in postnatal life. While short-term hot conditions (i.e. hot morning or hot chamber) consistently reduced nestling mitochondrial efficiency, prenatal exposure to heat-call altered this response, in the directions predicted by each of our hypotheses relating to the adaptive function of mitochondrial programming. Specifically, consistent with a reduction in heat production, exposure to prenatal heat-calls increased the mitochondrial respiration directed to ATP production (OXPHOS), thereby improving mitochondrial efficiency, across nest temperatures. Towards the hypothesis of oxidative damage reduction, *LEAK* respiration under extreme heat was higher in heat-call birds, and in faster growing nestlings (hence probably producing more ROS: [48]). No such correlation with growth rate was found under—much milder—in-nest conditions, where high *LEAK* (i.e. low mitochondrial efficiency) may thus not be beneficial. Taken together, our results demonstrate that mitochondrial programming by heat-calls could underlie differences in growth identified previously [40], with higher mitochondrial

**Table 1.** Results from reduced linear mixed models of the mitochondrial parameters as function of the playback (control-calls or heat-calls) and thermal conditions, with brood identity as random factor, in nestlings sampled under undisturbed in-nest conditions ($n = 20$) or after an experimental heat-challenge ($n = 19$). Thermal conditions include the mean daytime nest temperature experienced from hatching to 12 dph (12D-$T_{nest}$) and the nest temperature 3 h before sampling (AM-$T_{nest}$) for in-nest nestlings, or their deviation from the maximum temperature experienced in the chamber for heat-challenged nestlings ($\Delta$12D-$T_{nest}$, $\Delta$AM-$T_{nest}$). Est. = estimate, s.e. = standard error. Bold indicates significant effects ($p < 0.05$).

| response variable | fixed effect | Est. | s.e. | t | p-value |
|---|---|---|---|---|---|
| in-nest birds[a] | | | | | |
| ROUTINE | intercept | 3.41 | 0.20 | 17.31 | <0.001 |
| | AM-$T_{nest}$ | −0.39 | 0.19 | −1.98 | 0.063 |
| OXPHOS | intercept | 1.10 | 0.12 | 9.20 | <0.001 |
| | **playback (heat-call)** | **0.76** | **0.13** | **5.66** | **0.001** |
| | **AM-$T_{nest}$** | **−0.50** | **0.08** | **−6.27** | **<0.001** |
| LEAK | intercept | 1.96 | 0.13 | 15.45 | <0.001 |
| | 12D-$T_{nest}$ | 0.10 | 0.12 | 0.82 | 0.426 |
| ETS | intercept | 4.23 | 0.32 | 13.11 | <0.001 |
| | playback (heat-call) | 0.63 | 0.48 | 1.31 | 0.208 |
| FCR$_{L/R}$ | intercept | 0.64 | 0.02 | 27.05 | <0.001 |
| | **playback (heat-call)** | **−0.14** | **0.02** | **−6.17** | **0.001** |
| | **AM-$T_{nest}$** | **0.07** | **0.01** | **5.16** | **0.002** |
| FCR$_{L/ETS}$ | intercept | 0.49 | 0.02 | 21.45 | <0.001 |
| | **playback (heat-call)** | **−0.09** | **0.03** | **−2.70** | **0.015** |
| | 12D-$T_{nest}$ | 0.04 | 0.02 | 2.03 | 0.058 |
| FCR$_{R/ETS}$ | intercept | 0.78 | 0.03 | 24.92 | <0.001 |
| | 12D-$T_{nest}$ | 0.04 | 0.03 | 1.37 | 0.191 |
| heat-challenged birds[b] | | | | | |
| ROUTINE | intercept | 3.46 | 0.27 | 12.65 | <0.001 |
| | $\Delta$AM-$T_{nest}$ | 0.47 | 0.28 | 1.68 | 0.112 |
| OXPHOS | intercept | 1.65 | 0.16 | 10.58 | <0.001 |
| | $\Delta$AM-$T_{nest}$ | 0.21 | 0.16 | 1.29 | 0.214 |
| LEAK | intercept | 1.38 | 0.21 | 6.45 | <0.001 |
| | playback (heat-call) | 0.80 | 0.17 | 4.61 | 0.055 |
| | **$\Delta$12D-$T_{nest}$** | **0.49** | **0.19** | **2.60** | **0.020** |
| | $\Delta$AM-$T_{nest}$ | 0.36 | 0.19 | 1.92 | 0.080 |
| ETS | intercept | 4.18 | 0.25 | 16.87 | <0.001 |
| | $\Delta$12D-$T_{nest}$ | 0.37 | 0.25 | 1.47 | 0.161 |
| FCR$_{L/R}$ | intercept | 0.50 | 0.03 | 15.07 | <0.001 |
| | playback (heat-call) | 0.01 | 0.05 | 0.32 | 0.755 |
| | **$\Delta$12D-$T_{nest}$** | **0.09** | **0.03** | **2.84** | **0.012** |
| | **playback $\times$ $\Delta$12D-$T_{nest}$** | **−0.12** | **0.05** | **−2.65** | **0.018** |
| FCR$_{L/ETS}$ | intercept | 0.39 | 0.03 | 14.81 | <0.001 |
| | playback (heat-call) | 0.04 | 0.04 | 1.17 | 0.261 |
| | **$\Delta$12D-$T_{nest}$** | **0.06** | **0.03** | **2.36** | **0.034** |
| | **$\Delta$AM-$T_{nest}$** | **0.05** | **0.02** | **2.39** | **0.034** |
| | **playback $\times$ $\Delta$12D-$T_{nest}$** | **−0.12** | **0.04** | **−3.35** | **0.005** |
| FCR$_{R/ETS}$ | intercept | 0.82 | 0.03 | 24.15 | <0.001 |
| | $\Delta$AM-$T_{nest}$ | 0.07 | 0.03 | 1.90 | 0.074 |

[a]Full model: response $\sim$ playback + 12D-$T_{nest}$ + AM-$T_{nest}$ + playback $\times$ 12D-$T_{nest}$ + (1|brood id).
[b]Full model: response $\sim$ playback + $\Delta$12D-$T_{nest}$ + $\Delta$AM-$T_{nest}$ + playback $\times$ $\Delta$12D-$T_{nest}$ + (1|brood id).

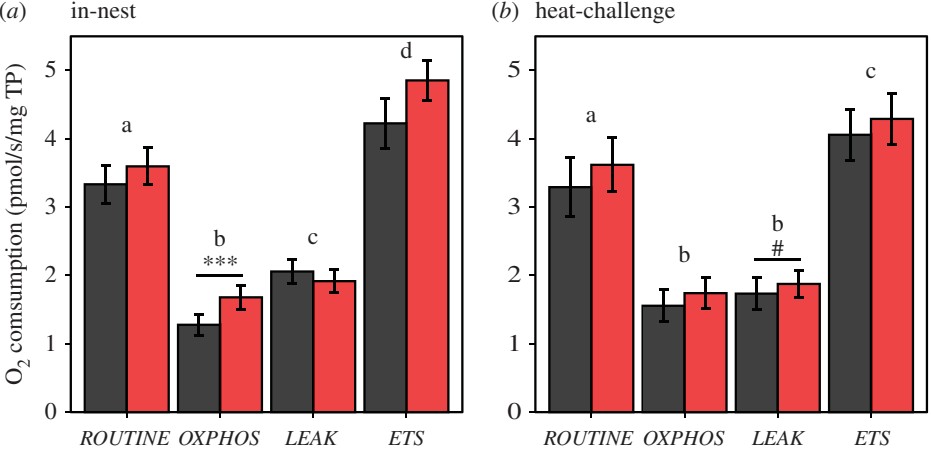

**Figure 1.** Average mitochondrial rates (±s.e.) in intact red blood cells from zebra finch nestlings exposed to prenatal playbacks of either heat-calls (red) or control-calls (black) and sampled following (a) in-nest (n, heat-calls = 9, control-calls = 11) or (b) acute heat-challenge conditions (n, heat-calls = 10, control-calls = 9). $O_2$ consumption rates are normalized by total protein content. Letters indicate significant differences ($p < 0.05$) between the mitochondrial rates (pooling both playback groups); ***$p = 0.001$, #$p = 0.055$ between playback groups. (Online version in colour.)

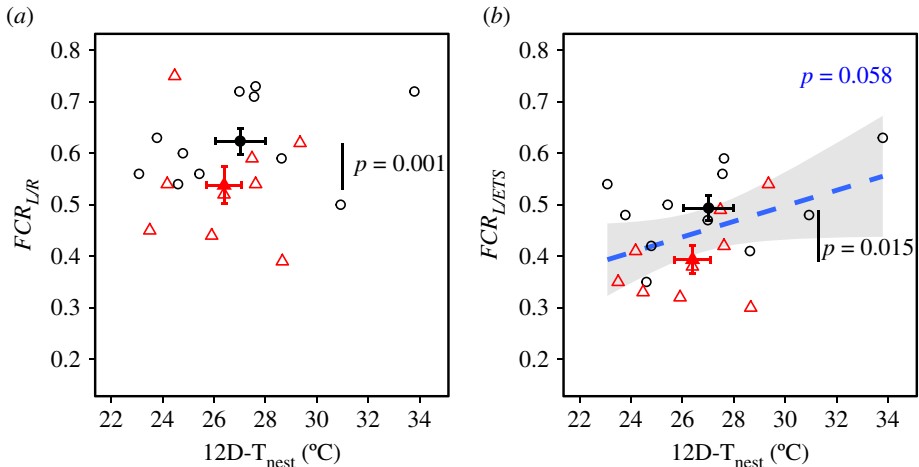

**Figure 2.** Mitochondrial efficiency in zebra finch nestlings in undisturbed in-nest conditions ($n = 20$), across the mean nest temperature experienced from hatching to 12dph (12D-$T_{nest}$). Birds were exposed prenatally to either control-calls (black circles) or heat-calls (red triangles). Mitochondrial efficiency was measured as *LEAK* relative to (a) respiration under endogenous conditions ($FCR_{L/R}$) or (b) mitochondria maximum working capacity ($FCR_{L/ETS}$); with lower ratio values indicating greater efficiency. Large solid circles show the means ± s.e. The regression line is represented with the 95% CIs. Removing the hottest nest in (a) and (b) did not change the results. (Online version in colour.)

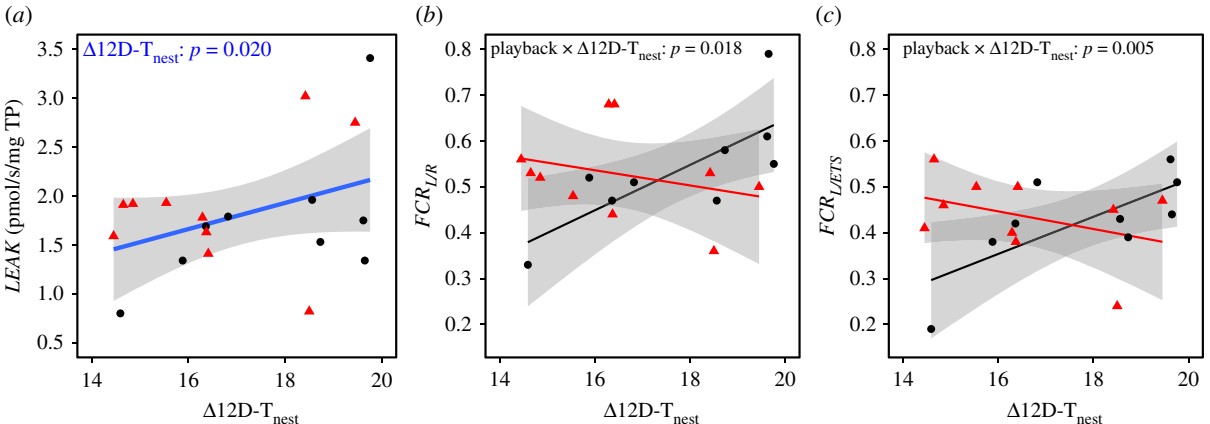

**Figure 3.** Mitochondrial parameters in zebra finch nestlings following an acute heat-challenge in a chamber ($n = 19$), across the chamber temperature above the mean nest temperature experienced from hatching to 12 dph (Δ12D-$T_{nest}$). Birds were exposed prenatally to either control-calls (black circles) or heat-calls (red triangles). (a) *LEAK* respiration; and mitochondrial efficiency measured as *LEAK* relative to (b) respiration under endogenous conditions ($FCR_{L/R}$) or (c) mitochondria maximum working capacity ($FCR_{L/ETS}$), with lower ratio values indicating greater efficiency. Regression lines are represented with the 95% CIs. (Online version in colour.)

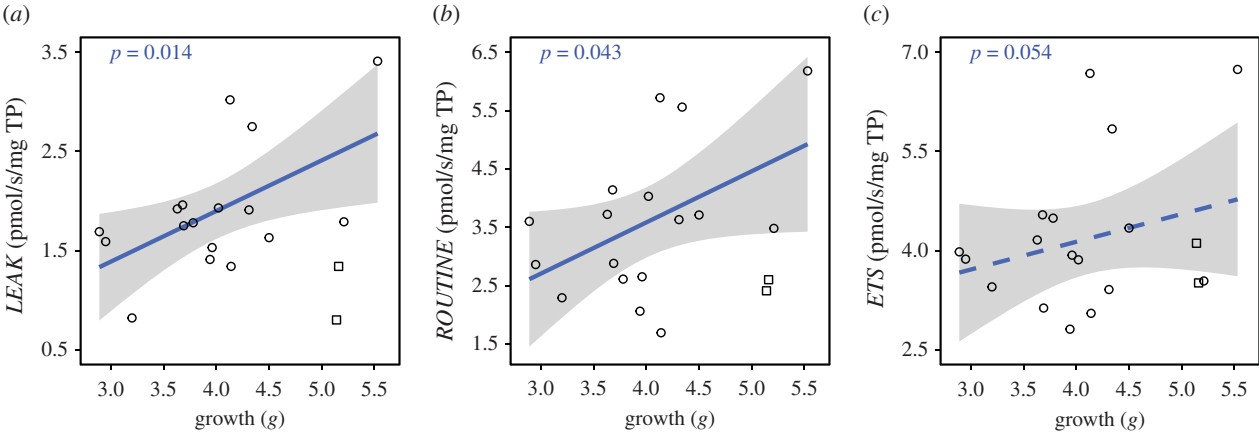

**Figure 4.** Relationship between growth rate (mass gain between day 7 and 12) of zebra finch nestlings exposed to an acute heat-challenge ($n = 19$) and (a) LEAK, (b) ROUTINE and (c) ETS respiration, normalized for protein content. Open squares show the two nestlings in broods of two. Excluding these two chicks did not change any of the results. Regression lines are shown with 95% CIs and correspond to all data points. (Online version in colour.)

efficiency under mild heat conditions expected to improve growth, and high LEAK under extreme heat to reduce it. Our findings therefore provide a mechanistic pathway for the adaptive programming of nestling development by prenatal heat-calls. Overall, our study reveals that prenatal sound alone is capable of reprogramming mitochondrial function, and provides a first line of evidence towards the adaptive function of prenatal mitochondrial programming.

Prenatal exposure to heat-calls induced a shift of nestling mitochondrial function towards higher ATP production (OXPHOS) and therefore lower $FCR_{L/R}$ and $FCR_{L/ETS}$ under undisturbed in-nest conditions. The prenatal acoustic signal, therefore, had similar impact on mitochondria to that of direct exposure to environmental temperatures in other species [26,28,31]. For example, in adults of three tit species, acclimatization to low winter temperatures increased LEAK respiration in RBCs [28]. Remarkably, in our study, effects of prenatal acoustic experience were observed two weeks post-exposure (13 dph), at the end of the nestling period. As a small altricial species, zebra finches are ectothermic at hatching (i.e. no endogenous heat production) and reach endothermy by about 10 dph [49]. Emergence of mitochondrial heat production is expected to precede endothermy acquisition, as in altricial red-winged blackbird nestlings (*Agelaius phoeniceus*), where LEAK and OXPHOS respiration start to increase at 3 dph [49]. In our experiment, the signature of heat-call exposure therefore persisted past the maturation of mitochondrial function for endothermy development. This suggests that changes to mitochondrial function might be long-lived, consistent with biomedical evidence documenting the long-term programming of mitochondrial function during prenatal life [11]. In wildlife, we are aware of only one study on the prenatal programming of mitochondrial function: chronic embryonic exposure to hypoxia reduced LEAK respiration and increased mitochondrial efficiency in 3-year-old juvenile alligators (*Alligator mississippiensis*) [50]. Our study therefore adds to this critical field by showing, for the first time, that adaptive mitochondrial programming also occurs in endotherms, even past the major mitochondrial remodelling underlying endothermy ontogeny.

In a previous study, we demonstrated that prenatal heat-call exposure adaptively increased subsequent nestling growth in cool shady nests, and reduced it in sunny nests with more severe increase in temperature (above ambient [40]). The higher OXPHOS and efficiency (as $FCR_{L/R}$ or $FCR_{L/ETS}$) in heat-call nestlings that we reveal here in in-nest conditions, would promote higher growth, as generally observed in both endotherms [51] and ectotherms [13,52,53]. All nestlings, including in hot nests, were sampled in mild heat conditions (less than 28°C), in late morning (11.30–12.45), before the peak of the heat (typically around 16.00). Our results therefore provide a mechanism for the previously demonstrated improved growth in heat-call birds reared in cool nests. Nonetheless, as nest temperatures increase throughout the day—particularly in hot nests—individual mitochondrial function also vary. When such conditions were simulated in the heated-chamber, LEAK was marginally higher ($p = 0.055$) in heat-call birds, and efficiency no longer higher than in control-call birds, thereby providing a mechanism contributing to reduced growth in heat-call nestlings in hot nests (as previously reported [40]). Furthermore, even though it was not measured here, it is possible that variation in food intake between heat- and control-call birds at high temperatures, would further emphasize growth differences [54]. Importantly, the effects on growth in this study did not match those described above, because our nestlings (from a mixture of cool and hot nests) were not reared solely in the conditions in which their mitochondrial function was tested (in mild and high heat). Nestling growth from day 7 to 12 therefore did not correlate with individuals' late morning mitochondrial profile in in-nest conditions. Nonetheless, at high temperature extremes, faster growing nestlings (across nest temperatures and playback groups) increased LEAK more (for oxidative damage reduction), suggesting that inter-individual growth rate differences could be evened up, if hot conditions persisted.

We found that hot conditions always reduced mitochondrial efficiency, regardless of the temporal scale or temperature range we considered, with average nest acclimation temperature, morning nest temperature or acute heat-challenge. The effect of average nest temperature, was not as pronounced, but in the same direction as acute responses, which may be indicative of a detrimental effect of high average temperatures, rather than an acclimation response. At high temperature extremes, however, increasing LEAK was probably an adaptive response, to reduce oxidative damage under heat-stress. Nonetheless, it is important to note that in addition to experiencing high temperatures, individuals in the heated-chamber were also food deprived, and placed in a novel environment. These additional stressors may have also contributed to the observed response. However, while the novel environment did not appear to affect the nestlings (sleeping in the dark chamber soon after

introduction), fasting mimics conditions nestlings would naturally experience when parents forgo provisioning under extreme heat [55,56]. Furthermore, the hot conditions used in our study, and their negative effect on mitochondrial efficiency, probably contributed to increased *LEAK* relative to total respiration (56% of *ROUTINE* on average, ranging from 33% to 79%). While some other published studies in passerine RBCs have observed similar values [15,21,28,37], future studies may establish whether *LEAK* is higher in desert-adapted avian species, and in developing young during fast growth phases. Lastly, it is noteworthy that all of the responses to temperature fluctuations that we report were observed in RBCs, as in some previous studies [21,28]. Even though mitochondrial function in RBCs correlates moderately with that in other tissues, including skeletal muscle [45,57], it is possible that effects would be more pronounced in more metabolically active tissues.

To conclude, our study demonstrates that adaptive developmental programming of mitochondrial function can occur via a prenatal acoustic signal. This aligns with recent evidence showing that sound may have a much larger impact on development and physiology than previously considered [6]. Our study also adds to the emerging literature on the modulations of mitochondrial function by environmental temperatures at multiple time scales [21,28,36]. Our findings therefore open exciting research avenues on the developmental programming of mitochondrial functions and heat adaptation in endotherms.

Ethics. All procedures described in this study were approved by the Animal Ethics Committee of Deakin University (G23–2018) and were carried out in accordance with the Australian guidelines and regulations for the use of animals in research.

Data accessibility. Data are available from the Dryad Digital Repository: https://doi.org/doi:10.5061/dryad.wm37pvmp7 [58]. The data are provided in electronic supplementary material [59].

Authors' contributions. E.U.: conceptualization, data curation, formal analysis, investigation, methodology, project administration, resources, software, validation, visualization, writing—original draft, writing—review and editing; J.M.G.: conceptualization, funding acquisition, methodology, resources, writing—review and editing; M.M.: conceptualization, funding acquisition, methodology, resources, writing—review and editing; A.P.: investigation, writing—review and editing; O.L.C.: funding acquisition, writing—review and editing; K.L.B.: conceptualization, funding acquisition, investigation, project administration, resources, writing—review and editing; M.M.M.: conceptualization, data curation, formal analysis, funding acquisition, investigation, methodology, project administration, resources, supervision, validation, visualization, writing—original draft, writing—review and editing. All authors gave final approval for publication and agreed to be held accountable for the work performed therein.

Competing interests. We declare we have no competing interests.

Funding. This work was supported by the Australian Research Council (grant nos. DP180101207 to K.L.B. and M.M.M., FT140100131 to K.L.B. and DE170100824 to M.M.M.), the British Biotechnology and Biological Sciences Research Council (BBSRC grant no. BB/S003223/1 to David F. Clayton, J.M.G., K.L.B. and M.M.M.) and Deakin University (internal grant to O.L.C. and M.M.).

Acknowledgements. We thank Nicolas de Almeida for help breeding the birds, and Rod Collins and the Deakin Animal House staff for their logistical support. We are grateful to two reviewers for their very useful comments.

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
