## [Peer Review File · Proceedings of the Royal Society B: Biological Sciences]

Review History

RSPB-2021-1893.R0 (Original submission)

Review form: Reviewer 1

Recommendation

Major revision is needed (please make suggestions in comments)

Scientific importance: Is the manuscript an original and important contribution to its field?
Excellent

General interest: Is the paper of sufficient general interest?
Excellent

Quality of the paper: Is the overall quality of the paper suitable?
Excellent

Is the length of the paper justified?
Yes

Should the paper be seen by a specialist statistical reviewer?
No

Do you have any concerns about statistical analyses in this paper? If so, please specify them explicitly in your report.

Yes

It is a condition of publication that authors make their supporting data, code and materials available - either as supplementary material or hosted in an external repository. Please rate, if applicable, the supporting data on the following criteria.

Is it accessible?

Yes

Is it clear?

Yes

Is it adequate?

Yes

Do you have any ethical concerns with this paper?

No

Comments to the Author

The study investigated whether the beneficial alterations observed in the growth of chicks exposed to parental heat calls (prenatal vocalizations emitted by parents when incubating at high temperatures), are mediated by changes in mitochondria bioenergetics. The experimental design allowed to test the effect of heat-related cues on mitochondria functions at three levels: prenatal heat call, acclimation nest temperature and acute heat stress. Specifically, nestlings were exposed to heat-calls (experimental) or contact-calls (controls) before hatching, while after hatching, they were acclimated to a certain nest temperature (gradient running from about 23 degrees to 34) experienced until day 12 post-hatching. In the third part of the study (heat stress experiment) some nestlings experienced a short-term (hours) heat stress (experimental) while the others were used as controls. In accordance with authors' expectations, the production of cellular heat (i.e. proton leak) was minimized when birds experienced pre-natal heat-calls or postnatal long-term heat challenges. Contrary to what expected, proton leak was not minimized when birds were exposed to an intense short-term heat-stress. Not expected was also the disruption of the beneficial response in producing less internal heat observed in both short or long heat challenges. Contrary to expectations, nestling growth rate was associated with proton leak and not with oxidative phosphorylation. This is the first study showing that mitochondria functions can be programmed by parental sound and open new perspectives on metabolic regulations. The study also represents an important contribution in understanding the role of mitochondria function during development at different temperatures. The observed response of mitochondria to external temperature can be applied to many other vertebrates making the paper of general interest. I have some comments that authors could find useful.

Models run by pooling data of 'heat-stress' and 'in-nest' chicks used to assess the effect of pre-natal heat calls in different conditions should be better explained. Specifically, the meaning of the interactive term "playback*12D-Tnest" used in models with pooled data is questionable (206) because in this dataset half of the birds were sampled after heat-stress and for these birds it is plausible to think that their physiology was not related to the acclimation conditions represented by the term "12D-Tnest" but to the heat-stress conditions. Consequently, the idea behind this statistical choice should be explained. In relation to this, the interaction term '12D-Tnest*experimental temperature' might be missing from the analysis because it is possible that the response to heat-stress could have been affected by acclimation conditions. This suggests that to understand the effect of pre-natal heat calls using this pooled dataset, the three-way interaction playback*12D-Tnest*experimental condition would be needed. Since this 3-way interaction term is difficult to interpret, I suggest to report only the more meaningful results related to the two

distinct datasets (also because the need of having models using full data set is not mentioned, 205).

The conditions experienced by controls in the short-term heat stress experiment differed from experimental birds not only in the exposure to high temperatures (lines 155-162) but also in other conditions. More specifically, “heat-challenged” birds were transferred and enclosed in a heated chamber for 2.5 hours and food-deprived, while controls remained in their known and safe nest, protected and fed by parents. A true control group would have been represented by birds exposed to fasting and that have been enclosed in the same chamber, but kept at room temperature. I think this is an important point because both fasting and psychological stress can activate the HPA axis with the release of corticosterone, which is able to induce mitochondria adjustments also in a relative short time (see the positive association between Cort and proton leak observed after 30 minutes from stressful event in Stier et al. 2019). Consequently, the effect of acute heat stress cannot be disentangled by fasting stress and nest/parent deprivation stress. This issue should be addressed in the discussion.

Overall, the formulation of expectations could be more detailed (107-123). Firstly, to better understand the expectations it would be important to specify how pre-natal heat calls ‘alter’ nestling development at high temperature (107). The cited study Mariette and Buchanan 2016 found a decrease in growth rate at high temperatures in chicks exposed to heat calls, while controls increased it. Then, it would be important to specify in which way this “mitochondrial programming” is expected to work (110) because knowing what found in Mariette and Buchanan 2016, a lower efficiency of mitochondria in producing ATP in birds exposed to heat calls and hotter nests should be expected, while the expectation reported here is the opposite (117): “we predicted that birds exposed to heat calls.....would minimize endogenous heat production by favoring ATP....” But if growth is supported by ATP/oxidative phosphorylation and if heat calls caused a decrease in growth rate, thus birds exposed to heat calls would minimize ATP. Consequently, it is not clear what is the rationale behind the formulation of the main prediction.

The overall expectation is that from a mitochondrial perspective the hotter means also the better (higher efficiency in producing ATP). Can this expectation be extended to other vertebrates or it is possible that species less adapted to high temperatures have mitochondria responding in a different way to heat? It would be nice to add some thoughts about the generality of these results, by making some considerations on the potential adaptation of zebra finches to heat and whether this adaptation could have influenced the observed results.

Other comments

The abstract could be a little bit more informative indicating the direction of the observed effects. i.e. (28): how was disrupted? (28-29): in which way mitochondria function was affected?

Methods:

146-147: it is not clear why temperature manipulation run from hatching to day 16 but the mean nest temperature was calculated excluding day 13-16.

169: “homogenate” means that cells have been mechanically micro-disrupted, which is not compatible with your protocol for intact cells. Please address this important point.

204: many chicks were siblings, especially in the pooled dataset, and the random factor “nest” should have been considered. However, since the factor “nest” was associated with a certain acclimation temperature, it was not possible to account for genetic/environmental similarities associated with “nest”, correct? A cross fostering design would have solved the problem. I think that authors should reassure the readers that this was not an important issue in their study?

209-213 and 267-273: the effects of mitochondrial trait on growth measured at day 12 (thus before heat stress experiment) was assessed in pooled dataset irrespective of the treatment (heat calls) or acclimation temperature. Can authors add some explanations on the reason why they did not include these terms in the model for growth rate? Was not important assessing the effect of heat calls on growth rate also when interacting with acclimation temperature to see whether the conditions of the 2016 study were present even here?

I could not find any description of how nestlings were allocated to the treatment and looking at the database did not help; e.g. sometimes all birds of one nest were allocated to the same treatment, in other case the two nests were split, etc. Please provide this info.

Result

Full results are missing, i.e. results obtained before and after dropping non-significant factors (procedure described in 217) should be presented at least in SEM together with AICc values (218) of compared models.

Discussion

291: it is not clear in which respect proton leak varies adaptively in nestlings.

Review form: Reviewer 2

Recommendation

Major revision is needed (please make suggestions in comments)

Scientific importance: Is the manuscript an original and important contribution to its field?

Excellent

General interest: Is the paper of sufficient general interest?

Good

Quality of the paper: Is the overall quality of the paper suitable?

Good

Is the length of the paper justified?

Yes

Should the paper be seen by a specialist statistical reviewer?

No

Do you have any concerns about statistical analyses in this paper? If so, please specify them explicitly in your report.

Yes

It is a condition of publication that authors make their supporting data, code and materials available - either as supplementary material or hosted in an external repository. Please rate, if applicable, the supporting data on the following criteria.

Is it accessible?

Yes

Is it clear?

No

Is it adequate?

No

Do you have any ethical concerns with this paper?

No

Comments to the Author

See PDF attached. (See Appendix A)

Decision letter (RSPB-2021-1893.R0)

30-Sep-2021

Dear Miss Udino:

Your manuscript has now been peer reviewed and the reviews have been assessed by an Associate Editor. The reviewers' comments (not including confidential comments to the Editor) and the comments from the Associate Editor are included at the end of this email for your reference. As you will see, the reviewers and the Editors have raised some concerns with your manuscript and we would like to invite you to revise your manuscript to address them.

Research ethics:

Use of animals and field studies:

It is a condition of publication that you make available the data and research materials supporting the results in the article. Please see our Data Sharing Policies (<https://royalsociety.org/journals/authors/author-guidelines/#data>). Datasets should be deposited in an appropriate publicly available repository and details of the associated accession number, link or DOI to the datasets must be included in the Data Accessibility section of the

article (<https://royalsociety.org/journals/ethics-policies/data-sharing-mining/>). Reference(s) to datasets should also be included in the reference list of the article with DOIs (where available).

[http://datadryad.org/submit?journalID=RSPB&manu=\(Document not available\)](http://datadryad.org/submit?journalID=RSPB&manu=(Document%20not%20available)), which will take you to your unique entry in the Dryad repository.

Please submit a copy of your revised paper within three weeks. If we do not hear from you within this time your manuscript will be rejected. If you are unable to meet this deadline please let us know as soon as possible, as we may be able to grant a short extension.

Best wishes,
Dr Locke Rowe
mailto: proceedingsb@royalsociety.org

Associate Editor

Comments to Author:

I have now received comments from two experts. They found the study is interesting and novel, but had major concerns on statistics, which make the conclusions of this study to be questionable. In addition, the reviewers questioned the methodology of mitochondrial function measurements, and the adaptive programming explanation of your results. Please pay attention on these major issues when you revise the paper.

Board Member: 2

Comments to Author(s):

Prenatal exposure to "heat-calls" adaptively alters nestling growth of the arid-adapted zebra finch. This is an interesting and remarkable finding. Extending their previous work, the authors determined the mitochondrial function of nestlings exposed to prenatal heat-call or not in the present paper. They show that prenatal heat-call can program mitochondrial function. The results not only identify a plausible mechanism for acoustic developmental programming, but also

highlight the role of prenatal sound (in addition to traditional cues like temperature) in developmental programming of mitochondrial function. Therefore, this paper brings us novel ideas and new knowledge on mitochondrial programming.

Reviewer(s)' Comments to Author:

Referee: 1

Comments to the Author(s)

The study investigated whether the beneficial alterations observed in the growth of chicks exposed to parental heat calls (prenatal vocalizations emitted by parents when incubating at high temperatures), are mediated by changes in mitochondria bioenergetics. The experimental design allowed to test the effect of heat-related cues on mitochondria functions at three levels: prenatal heat call, acclimation nest temperature and acute heat stress. Specifically, nestlings were exposed to heat-calls (experimental) or contact-calls (controls) before hatching, while after hatching, they were acclimated to a certain nest temperature (gradient running from about 23 degrees to 34) experienced until day 12 post-hatching. In the third part of the study (heat stress experiment) some nestlings experienced a short-term (hours) heat stress (experimental) while the others were used as controls. In accordance with authors' expectations, the production of cellular heat (i.e. proton leak) was minimized when birds experienced pre-natal heat-calls or postnatal long-term heat challenges. Contrary to what expected, proton leak was not minimized when birds were exposed to an intense short-term heat-stress. Not expected was also the disruption of the beneficial response in producing less internal heat observed in both short or long heat challenges. Contrary to expectations, nestling growth rate was associated with proton leak and not with oxidative phosphorylation. This is the first study showing that mitochondria functions can be programmed by parental sound and open new perspectives on metabolic regulations. The study also represents an important contribution in understanding the role of mitochondria function during development at different temperatures. The observed response of mitochondria to external temperature can be applied to many other vertebrates making the paper of general interest. I have some comments that authors could find useful.

Models run by pooling data of 'heat-stress' and 'in-nest' chicks used to assess the effect of pre-natal heat calls in different conditions should be better explained. Specifically, the meaning of the interactive term "playback*12D-Tnest" used in models with pooled data is questionable (206) because in this dataset half of the birds were sampled after heat-stress and for these birds it is plausible to think that their physiology was not related to the acclimation conditions represented by the term "12D-Tnest" but to the heat-stress conditions. Consequently, the idea behind this statistical choice should be explained. In relation to this, the interaction term '12D-Tnest*experimental temperature' might be missing from the analysis because it is possible that the response to heat-stress could have been affected by acclimation conditions. This suggests that to understand the effect of pre-natal heat calls using this pooled dataset, the three-way interaction playback*12D-Tnest*experimental condition would be needed. Since this 3-way interaction term is difficult to interpret, I suggest to report only the more meaningful results related to the two distinct datasets (also because the need of having models using full data set is not mentioned, 205).

The conditions experienced by controls in the short-term heat stress experiment differed from experimental birds not only in the exposure to high temperatures (lines 155-162) but also in other conditions. More specifically, "heat-challenged" birds were transferred and enclosed in a heated chamber for 2.5 hours and food-deprived, while controls remained in their known and safe nest, protected and fed by parents. A true control group would have been represented by birds exposed to fasting and that have been enclosed in the same chamber, but kept at room temperature. I think this is an important point because both fasting and psychological stress can activate the HPA axis with the release of corticosterone, which is able to induce mitochondria adjustments also in a relative short time (see the positive association between Cort and proton leak observed after 30 minutes from stressful event in Stier et al. 2019). Consequently, the effect of acute heat stress cannot be disentangled by fasting stress and nest/parent deprivation stress. This issue should be addressed in the discussion.

Overall, the formulation of expectations could be more detailed (107-123). Firstly, to better understand the expectations it would be important to specify how pre-natal heat calls 'alter' nestling development at high temperature (107). The cited study Mariette and Buchanan 2016 found a decrease in growth rate at high temperatures in chicks exposed to heat calls, while controls increased it. Then, it would be important to specify in which way this "mitochondrial programming" is expected to work (110) because knowing what found in Mariette and Buchanan 2016, a lower efficiency of mitochondria in producing ATP in birds exposed to heat calls and hotter nests should be expected, while the expectation reported here is the opposite (117): "we predicted that birds exposed to heat calls.....would minimize endogenous heat production by favoring ATP...." But if growth is supported by ATP/oxidative phosphorylation and if heat calls caused a decrease in growth rate, thus birds exposed to heat calls would minimize ATP. Consequently, it is not clear what is the rationale behind the formulation of the main prediction.

The overall expectation is that from a mitochondrial perspective the hotter means also the better (higher efficiency in producing ATP). Can this expectation be extended to other vertebrates or it is possible that species less adapted to high temperatures have mitochondria responding in a different way to heat? It would be nice to add some thoughts about the generality of these results, by making some considerations on the potential adaptation of zebra finches to heat and whether this adaptation could have influenced the observed results.

Other comments

The abstract could be a little bit more informative indicating the direction of the observed effects. i.e. (28): how was disrupted? (28-29): in which way mitochondria function was affected?

Methods:

146-147: it is not clear why temperature manipulation run from hatching to day 16 but the mean nest temperature was calculated excluding day 13-16.

169: "homogenate" means that cells have been mechanically micro-disrupted, which is not compatible with your protocol for intact cells. Please address this important point.

204: many chicks were siblings, especially in the pooled dataset, and the random factor "nest" should have been considered. However, since the factor "nest" was associated with a certain acclimation temperature, it was not possible to account for genetic/environmental similarities associated with "nest", correct? A cross fostering design would have solved the problem. I think that authors should reassure the readers that this was not an important issue in their study?

209-213 and 267-273: the effects of mitochondrial trait on growth measured at day 12 (thus before heat stress experiment) was assessed in pooled dataset irrespective of the treatment (heat calls) or acclimation temperature. Can authors add some explanations on the reason why they did not include these terms in the model for growth rate? Was not important assessing the effect of heat calls on growth rate also when interacting with acclimation temperature to see whether the conditions of the 2016 study were present even here?

I could not find any description of how nestlings were allocated to the treatment and looking at the database did not help; e.g. sometimes all birds of one nest were allocated to the same treatment, in other case the two nests were split, etc. Please provide this info.

Result

Full results are missing, i.e. results obtained before and after dropping non-significant factors (procedure described in 217) should be presented at least in SEM together with AICc values (218) of compared models.

Discussion

291: it is not clear in which respect proton leak varies adaptively in nestlings.

Referee: 2

Comments to the Author(s)

See PDF attached

Author's Response to Decision Letter for (RSPB-2021-1893.R0)

See Appendix B.

Decision letter (RSPB-2021-1893.R1)

15-Nov-2021

Dear Miss Udino

I am pleased to inform you that your manuscript entitled "Prenatal acoustic programming of mitochondrial function for high temperatures in an arid-adapted bird" has been accepted for publication in Proceedings B.

Data Accessibility section

Open Access

You are invited to opt for Open Access, making your freely available to all as soon as it is ready for publication under a CC BY licence. Our article processing charge for Open Access is £1700. Corresponding authors from member institutions (<http://royalsocietypublishing.org/site/librarians/allmembers.xhtml>) receive a 25% discount to these charges. For more information please visit <http://royalsocietypublishing.org/open-access>.

Paper charges

Sincerely,
Dr Locke Rowe
Editor, Proceedings B
[mailto: proceedingsb@royalsociety.org](mailto:proceedingsb@royalsociety.org)

Appendix A

This manuscript addresses an innovative and important topic: the programming of physiology by prenatal auditory cues. While prenatal programming of physiology by maternal diet, hormonal levels or incubation behaviour in birds has been fairly studied in the past decades, the potential for auditory cues to program postnatal physiology is really just emerging. The authors focused on cellular aerobic metabolism, which is increasingly considered as an important pathway underlying variation in individual performance and adaptation. The experimental design is solid (but N might be too low for appropriate statistics, see comment 1), the paper is well-written and I overall really enjoyed reading it. Yet, I have major concerns about the way the data is analysed, the choice of tissue (blood cells while animals were euthanized) and the quality of the mitochondrial measurements, which unfortunately could question the validity of the results and interpretations being presented.

I sincerely hope that my comments will be useful to the authors in revising their manuscript for *proc R Soc B* or another journal, and for the editor to reach a fair decision.

Best wishes

Dr. Antoine Stier [please note that I sign all my reviews]

Major comments:

1) **Non-independence of data points**

Birds used in this study are coming from non-independent data points as they share either biological (16 clutches? not crystal-clear from the dataset provided) and foster parents (12 nests of rearing? not crystal-clear from the dataset provided).

This is not taken into account in statistical models (by including random term(s)), which leads to pseudo-replication. Birds having the same biological parents are more likely to be similar in mitochondrial function than unrelated individuals since mitochondrial function is influenced by genetic background (e.g. Wolff et al. 2016 *J Evol Biol*). Similarly, birds sharing the same rearing environment are more likely to be similar in mitochondrial function than individuals reared in different conditions since mitochondrial function is likely influenced by early-life conditions (e.g. Casagrande et al. 2020 that you cite).

This is an issue for concluding about prenatal programming even if birds from a clutch / nest have been allocated to both control playback and heat-call playback, since in at least some clutches/broods, it seems that ≥ 2 nestlings share the same prenatal conditions.

This is even more problematic for testing postnatal nest temperature since all nestlings from a foster nest should have experienced the same postnatal temperature (but not only that, they also share the same parental care, etc..).

From the dataset provided, I am very surprised to see that two nestlings from the same nest do not share the same nest temperature. I can potentially understand it for the temperature 3 hours before sampling if timing of sampling differed between chicks within a nest, but I do not really understand how this could happen for the average temperature experienced during the 12 days of postnatal development. This issue should be clarified.

Consequently, I am afraid (and really sorry to say) that the statistics used here are likely not appropriate (linear models instead of linear mixed model), and that the limited sample size might not always enable the appropriate mixed models to converge (I tried quickly in SPSS and at least some models failed).

Taking at least into account the nest of rearing as a random factor would be of the highest importance in my opinion, especially considering the facts that postnatal temperature is similar within a nest and that I have unpublished data (Marciau & Stier, see poster at: https://www.researchgate.net/publication/336891355_Age_and_environment_but_not_genetics_affect_mitochondrial_function_in_wild_bird_species) from a cross-fostering experiment showing that the nest of rearing explains ~ 30% of variance in mitochondrial respiration rates, while the nest of origin explains very little.

2) Splitting analyses and then testing the overall dataset + multiple testing

I have some concerns about the statistical approach used by the authors, since they first analyse two subsets of data from the same overall experiment separately, and then analyse them together.

The appropriate procedure to the best of my knowledge would be the opposite, first testing in the general dataset if there is an effect of the playback, of the heat challenge and of their interaction. Only if there is a significant interaction, then splitting the dataset in two to test the effect of the playback or the heat-challenge separately would make sense, although post-hoc tests on interaction without splitting dataset should be possible in R using the emmeans package I think.

Following such approach, there is less (but still some!) support for a programming of mitochondrial bioenergetics by prenatal sound (i.e. significant playback effect on OXPHOS, marginally significant effect on FCR_L/R and FCR_L/ETS), but absolutely no evidence for an interaction between prenatal sound and postnatal acute heat-exposure. Yet, there is still the issue of pseudo-replication highlighted above that remains to be dealt with.

3) Testing relationship with fitness?

You mention lines 121-123: *“to understand the fitness consequences of mitochondrial function adjustments, we tested whether nestling growth rate was positively associated with mitochondrial efficiency or other mitochondrial parameters”*.

I do not really see how such approach could test for fitness consequences of mitochondrial function adjustment. What you test is an overall correlative link between growth rate (is it really a fitness proxy?) and mitochondrial traits (not adjustments), while not taking into account any experimental treatments (which itself could be problematic). It is impossible to tell from such correlative link if a particular mitochondrial phenotype causes differences in growth, if differences in growth causally influence mitochondrial phenotype, or if the two are linked through another non-measured parameter (e.g. food availability).

Consequently, I would suggest to remove this part from the manuscript as it does not add really meaningful information to your main story (prenatal acoustic programming) in my opinion.

4) The choice of blood cells to measure mitochondrial function

I have concerns that blood cells might not have been the best “tissue” to answer the scientific question presented in this manuscript.

It is clear that I am not against the use of blood cells to measure mitochondrial function, but in my opinion blood cells are mostly adequate in two scenarios: 1) you want repeated measurement of mitochondrial function through time from the same individual (e.g. Stier et al. 2019 Biol Lett), 2) you cannot kill your study animal for ethical (e.g. most wild animals, Stier et al. 2017) or scientific (e.g. obtaining subsequent data on performance, Ton et al. 2021) reasons. In the context of this study, zebra finches were in any case euthanized (line 164), which was thus allowing to collect tissues more relevant for heat tolerance than blood cells (e.g. skeletal muscle). Why did you choose to use blood cells instead of another tissue in this study?

Blood cells have a relatively low metabolic activity compared to muscle or liver (e.g. mitochondrial density is ~ 30/60 times lower in Japanese quail blood cells than muscle/liver; Stier et al. unpublished), and their contribution to endogenous heat production at the organismal level is therefore minimal (I made some quick calculations when Nord et al. FASEB 2021 paper was published, and the contribution of blood cells was < 1% of metabolic rate).

The authors should at least acknowledge that while mitochondrial function in blood cells is moderately correlated to other tissues, other more metabolically active tissues such as skeletal muscle would have been more appropriate to answer the study question.

5) High proton leak: biological or methodological artefact?

The values of proton leak respiration reported in this manuscript (>50% of endogenous respiration for in-nest and ~50% for heat-challenged birds) might appear as non-biological in my opinion and experience. Mitochondria “wasting” 50% of substrates (= food) not to produce ATP but to dissipate heat seems unlikely except in the case of a thermogenic tissue (e.g. brown adipose tissue of small mammals).

In my experience of working with avian red blood cells in various species, I have never (myself doing the labwork or closely supervising) obtained LEAK that high, i.e. mean = 56%, range = [33-79%]. Here are some published (and unfortunately a lot of unpublished) values as a comparison:

- King penguin (adults): 28% [19-38%] (Stier et al. 2017)
- Pied flycatcher (adults): 17% [13-21%] (Stier et al. 2019)
- Pied flycatcher (chicks + adults) : 25% [10-41%] (Marciau & Stier, unpublished)
- Japanese quail (chicks + adults): 15% [4-25%] (Stier et al. submitted to FASEB)
- Great tit (chicks): 23% [14-32%] (Cossin-Sevrin & Stier, submitted to J Exp Biol)

- Zebra finch (chicks + adults): 25% [16-38%] (Stier & Monaghan, unpublished)
- 25 bird species (adults): 17% [10-30%] (Stier et al. unpublished)

People I collaborated with (but was not on site to closely supervise data collection) have sometimes obtained somewhat higher LEAK values:

- Great tit (chicks) : 30% [6-52%] (Casagrande et al. 2020)
- Zebra finch (juveniles): 38% [28-54%] (Ton et al. 2021)
- Collared flycatcher (chicks + adults): 19% [13-26%] (Zhu & Qvarnström, unpublished)
- Eastern Yellow Robin (adults): 24% [10-53%] (Sunnucks & Stier, unpublished)

People using the technique without collaboration also tended to have high LEAK values:

- Blue tit (adults): autumn 38% [15-82%] / winter 54% [1-98%] (Nord et al. 2021)
- Coal tit (adults) : autumn 43% [21-65%] / winter 53% [29-82%] (Nord et al. 2021)
- Great tit (adults): autumn 17% [1-52%] / winter 36% [19-68%] (Nord et al. 2021)
- Zebra finch (adults): young ~47% [30-75%] / old ~35% [17-48%] (Dawson & Salmon 2020)

Consequently, I am wondering if such high *LEAK* values are really biological, or more stemming from issues in the laboratory protocol. In Divakaruni & Brand 2011 you cite, usual proton leak is considered ~ 20%, and leak of 50+% are considered remarkable exceptions or potential methodological artefacts. This could potentially affect the validity of the results obtained (to an unknown extent), and gives a wrong idea to the reader without expertise in mitochondrial biology of the potential magnitude of the proton leak phenomenon.

I acknowledge that it is the data you currently have and there is no easy way to tease apart biology from methodological issue here, but this limitation should at least be mentioned and acknowledged in the manuscript.

6) Adaptive programming of mitochondrial function?

I am not sure that you are really able to test the adaptive nature of the programming of mitochondrial function by prenatal sound (what is stated/implied line 104, 293, 313-315, 342-343 for instance) with this study. To this aim, you should be able to provide evidence that mitochondrial changes in response to prenatal heat-call (vs. control playback) are improving individual response to heat-stress/challenge. This would involve for instance blocking this mitochondrial response (e.g. by increasing proton leak with DNP?), measuring the response of control vs. heat-call playback birds (control vs. DNP) to heat challenge (e.g. body T°C change /body mass loss / water loss) and showing that the mitochondrial changes due to heat-call playback indeed drive the enhanced performance in response to heat challenge. This would be quite tricky (but elegant) to perform (and not sure DNP would be the right choice), but would be the proper way to prove adaptive programming through mitochondrial changes in my opinion.

Your data suggests that mitochondrial changes (i.e. higher efficiency due to higher OXPHOS, not lower LEAK) might be adaptive at high temperature (i.e. less heat produced per unit of ATP produced), but this remains to be really tested in my opinion. Consequently, I would

suggest to be more careful in your phrasing about adaptive programming all along the manuscript.

Other comments:

Nomenclature: I was told previously that it should be *ROUTINE* and *LEAK* instead of ROUTINE and LEAK, and that other mito parameters (ETS, OXPHOS, FCR) are supposed to be italicized as well (e.g. to avoid confusion between OXPHOS = the process of oxidative phosphorylation vs. *OXPHOS* = rate of respiration under phosphorylating conditions).

Lines 26-27: Prenatal heat-call did not shifted away respiration from proton leak since LEAK respiration was not affected by your prenatal playback, while OXPHOS was stimulated. Please reformulate

Lines 27-28 : In my opinion, you cannot say that since there is no statistical support for a disruption of the impact of prenatal heat calls on mitochondrial function by post-natal heat challenge, i.e. no significant playback*heat-challenge interaction.

Line 41: early-life instead of early?

Line 50: mitochondrial function instead of mitochondria?

Lines 48-50 + lines 62-63: the fact that it is unknown in animal is not fully correct since as you mentioned later on that Noguera et al. showed that prenatal acoustic cues could influence mitochondrial density (mtDNA copy number). This should be mentioned here as a first hint toward the prenatal programming of mitochondrial phenotype by acoustic cues.

Lines 94-95: the fact that proton leak is an essential source of heat production in birds has never been fully proven to the best of my knowledge, maybe rephrase to be less assertive?

Lines 119-121: there is no real direction in this prediction, so is it really useful?

Line 164: decapitated instead of decapitation?

Lines 167 & 189 : you analysed 46 samples but present data for 41 individuals. The reason for excluding these 5 samples should be explained in my opinion.

Lines 166-168 : what T°C have you used for measuring mitochondrial respiration?

Lines 212: maybe specify 10% of what (total N I guess)

Results : please be careful to always specify “was not significantly different” for all $p > 0.05$ (or 0.10 since you discuss ‘marginal’ effects). It is important since we cannot say there is no effect based on $p\text{-val} > 0.05$

Lines 282-284 : this would even argue against the adaptive nature of the prenatal programming, no?

Lines 285-286 : your more robust test for short-term effects of ambient temperature is the controlled heat-challenge you performed, and this had no significant effect on any of the mitochondrial parameters. This should be highlighted here in my opinion.

Line 293 : as mentioned in major comment 6, I would rephrase to be more cautious here.

Line 297 : mitochondria do not generate energy. They convert energy from one source (food) to other forms of energy (ATP + heat).

Line 310: neither your study nor the alligator one are wildlife studies since they were conducted on captive animals.

Lines 313-315: please rephrase according to major comment 6.

Lines 317-319: exposure to acute heat challenge had no significant effect, so you cannot state that in my opinion.

Lines 319-321: exposure to acute heat challenge had no significant effect, so you cannot state that in my opinion.

Lines 333-335: I don't think that you can state that [15] found a positive association between ATP production efficiency and growth performance. This paper reports the effects of oral CORT on both growth and mitochondrial function, but not on the relationship between growth and mitochondrial function.

Lines 335-338: in my opinion it is maybe too speculative to imply that.

Line 342: please rephrase according to major comment 6

References (not cited in the manuscript):

Wolff, J. N. et al. Evolutionary implications of mitochondrial genetic variation: mitochondrial genetic effects on OXPHOS respiration and mitochondrial quantity change with age and sex in fruit flies. Journal of Evolutionary Biology 29, 736–747 (2016).

Appendix B

Associate Editor

I have now received comments from two experts. They found the study is interesting and novel, but had major concerns on statistics, which make the conclusions of this study to be questionable. In addition, the reviewers questioned the methodology of mitochondrial function measurements, and the adaptive programming explanation of your results. Please pay attention on these major issues when you revise the paper.

>> We thank the Associate Editor for giving us the opportunity to revise our paper. We have now explained the statistics better and corrected them as needed. We have also cleared any doubts over our methodology and clarified the adaptive significance of our findings.

Board Member: 2

Prenatal exposure to “heat-calls” adaptively alters nestling growth of the arid-adapted zebra finch. This is an interesting and remarkable finding. Extending their previous work, the authors determined the mitochondrial function of nestlings exposed to prenatal heat-call or not in the present paper. They show that prenatal heat-call can program mitochondrial function. The results not only identify a plausible mechanism for acoustic developmental programming, but also highlight the role of prenatal sound (in addition to traditional cues like temperature) in developmental programming of mitochondrial function. Therefore, this paper brings us novel ideas and new knowledge on mitochondrial programming.

>> We thank the Board Member for their positive assessment of our study and for recognising its novelty and contribution to the fields of prenatal communication and mitochondrial programming.

Referee: 1

The study investigated whether the beneficial alterations observed in the growth of chicks exposed to parental heat calls (prenatal vocalizations emitted by parents when incubating at high temperatures), are mediated by changes in mitochondria bioenergetics. The experimental design allowed to test the effect of heat-related cues on mitochondria functions at three levels: prenatal heat call, acclimation nest temperature and acute heat stress. Specifically, nestlings were exposed to heat-calls (experimental) or contact-calls (controls) before hatching, while after hatching, they were acclimated to a certain nest temperature (gradient running from about 23 degrees to 34) experienced until day 12 post-hatching. In the third part of the study (heat stress experiment) some nestlings experienced a short-term (hours) heat stress (experimental) while the others were used as controls. In accordance with authors' expectations, the production of cellular heat (i.e. proton leak) was minimized when birds experienced pre-natal heat-calls or postnatal long-term heat challenges. Contrary to what expected, proton leak was not minimized when birds were exposed to an intense short-term heat-stress. Not expected was also the disruption of the beneficial response in producing less internal heat observed in both short or long heat challenges. Contrary to expectations, nestling growth rate was associated with proton leak and not with oxidative phosphorylation. This is the first study showing that mitochondria functions can be programmed by parental sound and open new perspectives on metabolic regulations. The study also represents an important contribution in understanding the role of mitochondria function during development at different temperatures. The observed response of

mitochondria to external temperature can be applied to many other vertebrates making the paper of general interest. I have some comments that authors could find useful.

>> We thank Reviewer 1 for their positive assessment of our study and its contribution to the field. We are also very grateful for the Reviewer's extremely useful comments below.

Models run by pooling data of 'heat-stress' and 'in-nest' chicks used to assess the effect of pre-natal heat calls in different conditions should be better explained. Specifically, the meaning of the interactive term "playback*12D-Tnest" used in models with pooled data is questionable (206) because in this dataset half of the birds were sampled after heat-stress and for these birds it is plausible to think that their physiology was not related to the acclimation conditions represented by the term "12D-Tnest" but to the heat-stress conditions. Consequently, the idea behind this statistical choice should be explained. In relation to this, the interaction term '12D-Tnest*experimental temperature' might be missing from the analysis because it is possible that the response to heat-stress could have been affected by acclimation conditions. This suggests that to understand the effect of pre-natal heat calls using this pooled dataset, the three-way interaction playback*12D-Tnest*experimental condition would be needed. Since this 3-way interaction term is difficult to interpret, I suggest to report only the more meaningful results related to the two distinct datasets (also because the need of having models using full data set is not mentioned, 205).

>> As recommended by the reviewer, we now only focus on analysing the two distinct datasets separately, to avoid confusion and difficulty over the interpretation of the combined dataset. As the Reviewer also rightly pointed out in their next comment (see below), the several differences between in-nest and heat-challenge birds may preclude pooling them together.

We also agree that the effect of 12D-Tnest after a heat challenge may not be comparable to its effects in the in-nest conditions, and thank the reviewer for pointing this out. To avoid misinterpretation of nest temperature after a heat challenge, and directly test whether "the response to heat-stress could have been affected by acclimation conditions" as recommended by the reviewer, in the analyses for heat-challenged birds, we have now replaced 12D-Tnest by a new predictor representing the deviation between the heated-chamber temperature and 12D-Tnest. This predictor therefore measures how extreme the heated-chamber conditions were, compared to the individual's acclimation temperature in the nest. Changing predictors (which simply equates to adding a constant of the maximal chamber temperature to all 12D-Tnest values) did not change the results, but greatly facilitated their interpretation: there was still a significant interaction of playback and nest temperature deviation on mitochondrial efficiency (for FCRs L/R and L/ETS), but we showed that, control-call birds from the heated-chamber followed the same pattern as for in-nest conditions, with efficiency decreasing in the relatively hotter chamber, whereas this was not the case for the heat-calls birds. We updated the predictions (L161-173, with tracked changes), methods (L332-374), and results (L378-462) accordingly.

For the reasons highlighted above, and given mitochondrial traits may have a different significance meaning under undisturbed and heat-challenged conditions, we also now run growth analyses for the two experiments separately.

The conditions experienced by controls in the short-term heat stress experiment differed from experimental birds not only in the exposure to high temperatures (lines 155-162) but also in other conditions. More specifically, "heat-challenged" birds were transferred and enclosed in a heated chamber for 2.5 hours and food-deprived, while controls remained in their known and safe nest, protected and fed by parents. A true control group would have been represented by birds exposed to

fasting and that have been enclosed in the same chamber, but kept at room temperature. I think this is an important point because both fasting and psychological stress can activate the HPA axis with the release of corticosterone, which is able to induce mitochondria adjustments also in a relative short time (see the positive association between Cort and proton leak observed after 30 minutes from stressful event in Stier et al. 2019). Consequently, the effect of acute heat stress cannot be disentangled by fasting stress and nest/parent deprivation stress. This issue should be addressed in the discussion.

>> We acknowledge the point raised by the reviewer, and agree in-nest and heat-challenged birds varied by more than one factor. By testing individuals that were in their nests (in-nest conditions), we aimed to first investigate the effects of acoustic experience and acclimation temperature on mitochondrial function under undisturbed conditions, which had never been investigated before. The heat-challenged birds indeed experienced more than thermal stress, as they were also food-deprived, and placed in a novel environment. The novel environment did not appear to affect the nestlings, which were sleeping in the dark chamber soon after they were introduced. Fasting may indeed cause additional stress to the individual in addition to the heat-challenge, but mimics conditions birds would naturally experience when parents forgo provisioning under extreme heat. This is why we chose to combine fasting and heat challenge. As above, by analysing these datasets separately any confounds are negated and we now mention the potential effect of these additional factors in the discussion (L673-678).

Overall, the formulation of expectations could be more detailed (107-123). Firstly, to better understand the expectations it would be important to specify how pre-natal heat calls 'alter' nestling development at high temperature (107). The cited study Mariette and Buchanan 2016 found a decrease in growth rate at high temperatures in chicks exposed to heat calls, while controls increased it. Then, it would be important to specify in which way this "mitochondrial programming" is expected to work (110) because knowing what found in Mariette and Buchanan 2016, a lower efficiency of mitochondria in producing ATP in birds exposed to heat calls and hotter nests should be expected, while the expectation reported here is the opposite (117): "we predicted that birds exposed to heat calls.....would minimize endogenous heat production by favoring ATP...." But if growth is supported by ATP/oxidative phosphorylation and if heat calls caused a decrease in growth rate, thus birds exposed to heat calls would minimize ATP. Consequently, it is not clear what is the rationale behind the formulation of the main prediction.

>> We thank the reviewer for drawing our attention to this ambiguity. We now describe the specific growth patterns observed in Mariette and Buchanan 2016 in relation to nest temperature (L129-130), and explain why we predicted a higher mitochondrial efficiency in heat-call birds (L161-166).

We expected the prenatal reprogramming of mitochondrial function by heat-calls (forecasting hot postnatal conditions) to increase efficiency based on the advantage of reducing heat production in hot environments, rather than specifically for explaining growth, which also varies with food intake (not measured). Higher mitochondrial efficiency alone would not suffice in explaining lower growth in hot conditions, but, importantly, is compatible with the differential growth patterns observed in Mariette and Buchanan 2016. We absolutely agree that a higher mitochondrial efficiency would lead to higher growth; but only when food intake is constant. Yet, food intake in both birds and mammals, is known to decrease in hot conditions, to reduce the unavoidable heat production associated with food digestion (i.e. heat increment of feeding). A high mitochondrial efficiency (producing more ATP per food unit) decreases the amount of food needed to be consumed to achieve a given growth. In

hot conditions, a high mitochondria efficiency may be adaptive as it reduces heat production, by decreasing necessary food intake, in addition to decreasing endogenous heat production associated with LEAK; lower growth may simply result from further reducing food intake. In cool conditions, when food intake is not inhibited by heat, a high mitochondrial efficiency could increase growth, as we previously observed in heat-call birds.

However, we now revised our predictions, to also include the possibility that decreased growth in heat-call birds in hot conditions could be due to an increase in LEAK (and therefore a decrease in mitochondrial efficiency). While higher LEAK would be expected to increase endogenous heat-production, it would also decrease oxidative damage. Indeed, in Mariette & Buchanan 2016, we suggested (without testing it) that heat-call exposure may reduce growth in the heat to limit the oxidative damage of growing in the heat, thereby leading to subsequent positive effects on reproductive success at adulthood.

Our clearer interpretation of the results in in-nest versus heat-challenge birds now show that both hypotheses are supported. However, higher LEAK in heat-call birds only occurs under extreme heat (as in the heat-challenge), rather than under the conditions encountered in hot nest conditions in late morning (11:30 to 12:45) before the peak of the heat (typically at 16:00). Indeed we found that under undisturbed nest conditions, mitochondrial efficiency mostly varies with morning temperature rather than with the average temperature nestlings experienced since hatching.

The overall expectation is that from a mitochondrial perspective the hotter means also the better (higher efficiency in producing ATP). Can this expectation be extended to other vertebrates or it is possible that species less adapted to high temperatures have mitochondria responding in a different way to heat? It would be nice to add some thoughts about the generality of these results, by making some considerations on the potential adaptation of zebra finches to heat and whether this adaptation could have influenced the observed results.

>> We agree with the reviewer that the response we observed may be specific to desert-adapted species, or hot conditions. High mitochondrial efficiency is beneficial for both limiting heat production and increasing growth. However, in cooler environments, mitochondrial efficiency needs to be trade-off against the need for heat production to maintain normal body temperature. Our rewording of predictions (L161-173) will undoubtedly make this clearer and we also mentioned the possibility that desert-adapted species may be particular in the discussion (L681-682).

Other comments

The abstract could be a little bit more informative indicating the direction of the observed effects. i.e. (28): how was disrupted? (28-29): in which way mitochondria function was affected?

>> We have reworded the abstract to provide more detailed results.

Methods:

146-147: it is not clear why temperature manipulation run from hatching to day 16 but the mean nest temperature was calculated excluding day 13-16.

>> The mean nest temperature was calculated until day 12 because individuals were tested on day 13, around midday. The nest temperature manipulation ran until day 16 as part of another study. However, given this manipulation between days 14 and 16 is irrelevant to this study, we now only refer to the nest manipulation until 13 days to prevent any confusion (L225). This should also clarify that nest temperature manipulation was still occurring on day 13 until testing.

169: “homogenate” means that cells have been mechanically micro-disrupted, which is not compatible with your protocol for intact cells. Please address this important point.

>> Thank you for pointing this out. We replaced “homogenate” by “resuspension”.

204: many chicks were siblings, especially in the pooled dataset, and the random factor “nest” should have been considered. However, since the factor “nest” was associated with a certain acclimation temperature, it was not possible to account for genetic/environmental similarities associated with “nest”, correct? A cross fostering design would have solved the problem. I think that authors should reassure the readers that this was not an important issue in their study?

>> We indeed used a cross-fostering design in our study (formerly lines 139-140). This is now indicated more clearly in the text (L220).

We had indeed included “nest identity” (representing brood identity) as a random factor in our models, since we agree it could, a priori, be an important source of variation. However, since the random effect for brood ID was null in many models, we excluded it from all analyses for parsimony, but unfortunately deleted the sentence saying so in the methods. We now corrected this by retaining Brood ID in models, unless the variance of this random factor was null. This is now explained in the statistical analyses section (L332) and we updated the results accordingly. It did not however change the results of our study.

209-213 and 267-273: the effects of mitochondrial trait on growth measured at day 12 (thus before heat stress experiment) was assessed in pooled dataset irrespective of the treatment (heat calls) or acclimation temperature. Can authors add some explanations on the reason why they did not include these terms in the model for growth rate? Was not important assessing the effect of heat calls on growth rate also when interacting with acclimation temperature to see whether the conditions of the 2016 study were present even here?

>> Our aim was to assess the relationship between mitochondrial traits and growth, irrespective of which developmental factors may have caused variation in growth (rather than after accounting for possible effects of developmental experience). Adding prenatal playback and nest temperature did not change the outcome of these analyses. We now nonetheless analyse in-nest and heat-challenge experiments separately (please see above).

I could not find any description of how nestlings were allocated to the treatment and looking at the database did not help; e.g. sometimes all birds of one nest were allocated to the same treatment, in other case the two nests were split, etc. Please provide this info.

>> Nestlings were allocated to the prenatal acoustic treatments on day 10 of incubation, when eggs from the same clutch were randomly split between the two playback groups (now specified L236-238).

Mitochondrial respiration was always tested when nestlings were 12 days old. Because of equipment limitations, we could only assess mitochondrial function for up to 2 nestlings per day, and put only one nestling through the heat-challenge. Each day, we therefore aimed to test two nestlings from the same playback group, with one in each of the two experimental conditions (nest or heat-challenge), except on 3 days when there were not two 12-day old nestlings from the same playback group available for testing. Within broods, nestlings were randomly allocated to either the in-nest or heat-challenge conditions, making sure there was no overall bias in nestling mass or hatching order. Nestlings were always raised in mixed broods of heat- and control-call individuals, but not all broodmates were necessarily tested in this experiment (because we could only test 2 (12-day old) nestlings per day). For broods with untested (and therefore surviving) nestlings, we avoided leaving only 1 live nestling, which may have led to nest abandonment.

We have now provided this information in the methods section (L238-240).

Result

Full results are missing, i.e. results obtained before and after dropping non-significant factors (procedure described in 217) should be presented at least in SEM together with AICc values (218) of compared models.

>> As recommended by the reviewer, we added tables with the full models for all analyses in the supplementary material, along with comparisons of the full and reduced models (tables S2, S3, S4).

Discussion

291: it is not clear in which respect proton leak varies adaptively in nestlings.

>> This sentence has now been deleted in the process of rewriting the discussion to clarify the adaptive values of the effects we observed (also now clearer with the reanalysis of the heat-challenge experiment).

Referee: 2

This manuscript addresses an innovative and important topic: the programming of physiology by prenatal auditory cues. While prenatal programming of physiology by maternal diet, hormonal levels or incubation behaviour in birds has been fairly studied in the past decades, the potential for auditory cues to program postnatal physiology is really just emerging. The authors focused on cellular aerobic metabolism, which is increasingly considered as an important pathway underlying variation in individual performance and adaptation. The experimental design is solid (but N might be too low for appropriate statistics, see comment 1), the paper is well-written and I overall really enjoyed reading it. Yet, I have major concerns about the way the data is analysed, the choice of tissue (blood cells while animals were euthanized) and the quality of the mitochondrial measurements, which unfortunately could question the validity of the results and interpretations being presented.

I sincerely hope that my comments will be useful to the authors in revising their manuscript for *Proc R Soc B* or another journal, and for the editor to reach a fair decision.

Best wishes

Dr. Antoine Stier [please note that I sign all my reviews]

>> We thank the Reviewer for acknowledging the importance and novelty of our findings and research topic. We have addressed all of the Reviewer's major comments, including the clarification and revision of our analyses, and the validation of our methodology and approach using red blood cells.

Major comments:

1) Non-independence of data points

Birds used in this study are coming from non-independent data points as they share either biological (16 clutches? not crystal-clear from the dataset provided) and foster parents (12 nests of rearing? not crystal-clear from the dataset provided).

This is not taken into account in statistical models (by including random term(s)), which lead to pseudo-replication. Birds having the same biological parents are more likely to be similar in mitochondrial function than unrelated individuals since mitochondrial function is influenced by genetic background (e.g. Wolff et al. 2016 J Evol Biol). Similarly, birds sharing the same rearing environment are more likely to be similar in mitochondrial function than individuals reared in different conditions since mitochondrial function is likely influenced by early-life conditions (e.g. Casagrande et al. 2020 that you cite).

This is an issue for concluding about prenatal programming even if birds from a clutch / nest have been allocated to both control playback and heat-call playback, since in at least some clutches/broods, it seems that ≥ 2 nestlings share the same prenatal conditions.

This is even more problematic for testing postnatal nest temperature since all nestlings from a foster nest should have experienced the same postnatal temperature (but not only that, they also share the same parental care, etc..).

From the dataset provided, I am very surprised to see that two nestlings from the same nest do not share the same nest temperature. I can potentially understand it for the temperature 3 hours before sampling if timing of sampling differed between chicks within a nest, but I do not really understand how this could happen for the average temperature experienced during the 12 days of postnatal development. This issue should be clarified.

Consequently, I am afraid (and really sorry to say) that the statistics used here are likely not appropriate (linear models instead of linear mixed model), and that the limited sample size might not always enable the appropriate mixed models to converge (I tried quickly in SPSS and at least some models failed).

Taking at least into account the nest of rearing as a random factor would be of the highest importance in my opinion, especially considering the facts that postnatal temperature is similar within a nest and that I have unpublished data (Marciau & Stier, see poster at: https://www.researchgate.net/publication/336891355_Age_and_environment_but_not_genetics_affect_mitochondrial_function_in_wild_bird_species) from a cross-fostering experiment showing that the nest of rearing explains $\sim 30\%$ of variance in mitochondrial respiration rates, while the nest of origin explains very little.

>> a. The reviewer strongly recommends considering the rearing nest as a random factor, as also suggested by reviewer 1.

Please also see our response to Reviewer 1 above. We can reassure the Reviewer that our analyses were appropriate, but admittedly, not sufficiently detailed: Brood ID as a random factor was omitted

from our analyses for parsimony because its variance is null in many cases (in which case the model does not consider it). This is now corrected (L332, also see supplementary tables S2-S3), which did not change our findings.

The variance in some models is null, not because of a lack of power, but in part because we are already accounting for some developmental effects in fixed predictors (so variation that would be attributed to “brood” when nest conditions are not measured could here be attributed to nest temperature). It is also likely to be null because most data points were actually independent, since, within experiments (i.e. in-nest conditions or heat-challenge) only some individuals had broodmates in the dataset: out of 15 or 14 different broods in in-nest and heat-challenged experiments respectively, only 4 included 2 or 3 individuals tested in this study.

We agree with the reviewer that while rearing nest may be important, the nest of origin (clutch identity) is less likely to explain any variation. We thank the Reviewer for providing a link to interesting unpublished work. For completeness, we also present tables in supplementary material with “clutch identity” as a random factor unless its variance was null. As for broods, most clutches only had 1 sibling tested per experiment. Out of 16 or 17 different clutches (for in-nest or heat-challenge experiments respectively), only 4 or 2 (respectively) included 2 individuals tested.

b. The reviewer seeks clarifications on the mean nest temperature values for broodmates

The mean nest temperature was calculated per individual, from hatching to day 12. In our dataset, nestlings reared in the same nest hatched 1, 2 or 3 days apart, which explains the slight variation in mean nest temperature experienced by each of the broodmates. Some broods also have similar names, which may have caused some confusion. Broods are now renamed with numbers in the dataset to avoid confusion and the calculation of mean nest temperature per individuals clarified in the text (L226-228).

c. The reviewer is concerned about the use of mixed models given the sample size

Using R and the package *lme4*, none of the linear mixed models (including nest identity as random factor) ran in this study had convergence failures, indicating that the sample size is sufficient for running our analyses.

2) **Splitting analyses and then testing the overall dataset + multiple testing**

I have some concerns about the statistical approach used by the authors, since they first analyse two subsets of data from the same overall experiment separately, and then analyse them together.

The appropriate procedure to the best of my knowledge would be the opposite, first testing in the general dataset if there is an effect of the playback, of the heat challenge and of their interaction. Only if there is a significant interaction, then splitting the dataset in two to test the effect of the playback or the heat-challenge separately would make sense, although post-hoc tests on interaction without splitting dataset should be possible in R using the emmeans package I think.

Following such approach, there is less (but still some!) support for a programming of mitochondrial bioenergetics by prenatal sound (i.e. significant playback effect on OXPHOS, marginally significant effect on FCR_L/R and FCR_L/ETS), but absolutely no evidence for an interaction between prenatal sound and postnatal acute heat-exposure. Yet, there is still the issue of pseudo-replication highlighted above that remains to be dealt with.

>> As recommended by Reviewer 1, the two datasets are not exactly comparable because the birds who experienced the heat-challenge were, in addition to heat, also exposed to other stressors (closed chamber and food deprivation) that the in-nest birds did not experience. Therefore, the birds from the in-nest condition are not a true control. Nest temperature also has a different interpretation depending on whether or not the birds were sampled from their nest or after a heat-challenge. Considering this, we agree with Reviewer 1 that the appropriate procedure as we follow now is to only perform analyses on for two distinct experiments separately.

Pseudo-replication does not exist within the analysis, as detailed above.

3) Testing relationship with fitness?

You mention lines 121-123: *“to understand the fitness consequences of mitochondrial function adjustments, we tested whether nestling growth rate was positively associated with mitochondrial efficiency or other mitochondrial parameters”*.

I do not really see how such approach could test for fitness consequences of mitochondrial function adjustment. What you test is an overall correlative link between growth rate (is it really a fitness proxy?) and mitochondrial traits (not adjustments), while not taking into account any experimental treatments (which itself could be problematic). It is impossible to tell from such correlative link if a particular mitochondrial phenotype causes differences in growth, if differences in growth causally influence mitochondrial phenotype, or if the two are linked through another non-measured parameter (e.g. food availability). Consequently, I would suggest to remove this part from the manuscript as it does not add really meaningful information to your main story (prenatal acoustic programming) in my opinion.

>> We have now clarified the purpose of these analyses on nestling growth, and included the developmental conditions (prenatal playback and mean nest temperature) in the analyses. These analyses are important for the interpretation of our results, and their adaptive values (see text), and were therefore retained. We agree that the directionality of these effects is unknown in principle; this does not however make them less valuable.

4) The choice of blood cells to measure mitochondrial function

I have concerns that blood cells might not have been the best “tissue” to answer the scientific question presented in this manuscript.

It is clear that I am not against the use of blood cells to measure mitochondrial function, but in my opinion blood cells are mostly adequate in two scenarios: 1) you want repeated measurement of mitochondrial function through time from the same individual (e.g. Stier et al. 2019 Biol Lett), 2) you cannot kill your study animal for ethical (e.g. most wild animals, Stier et al. 2017) or scientific (e.g. obtaining subsequent data on performance, Ton et al. 2021) reasons. In the context of this study, zebra finches were in any case euthanized (line 164), which was thus allowing to collect tissues more relevant for heat tolerance than blood cells (e.g. skeletal muscle). Why did you choose to use blood cells instead of another tissue in this study?

Blood cells have a relatively low metabolic activity compared to muscle or liver (e.g. mitochondrial density is ~ 30/60 times lower in Japanese quail blood cells than muscle/liver; Stier et al. unpublished), and their contribution to endogenous heat production at the organismal level is therefore minimal (I made some quick calculations when Nord et al. FASEB2021 paper was published, and the contribution of blood cells was < 1% of metabolic rate).

The authors should at least acknowledge that while mitochondrial function in blood cells is moderately correlated to other tissues, other more metabolically active tissues such as skeletal muscle would have been more appropriate to answer the study question.

>> In our experiment, nestlings were indeed sacrificed to collect tissues, which were used for the purpose of another study, and were not available for measuring mitochondrial function. However, that other tissues would have potentially been available does not make our findings less reliable. We agree with the benefits of using red blood cells to allow repeated measures or non-terminal sampling, but that our experiment would not correspond to these scenarios does not, in itself, make RBC any less suitable. Our study suffers from the same limitations as others using this same approach with RBC, which we now highlight in the discussion (L684-686).

Importantly, while we fully agree that other tissues than blood may show stronger effects, we disagree with the suggestion that RBC are not suitable to measure responses to thermal change: the study by Nord et al FASEB 2021 cited by the reviewer, did show that RBC mitochondrial function varies seasonally, with notably, LEAK being higher in winter than autumn, when higher endogenous heat production is required. Moreover, in Ton et al 2021, a study which the reviewer also mentioned and to which he collaborated, rearing nest temperatures affected zebra finch mitochondrial parameters in RBC.

5) **High proton leak: biological or methodological artefact?**

The values of proton leak respiration reported in this manuscript (>50% of endogenous respiration for in-nest and ~50% for heat-challenged birds) might appear as non-biological in my opinion and experience. Mitochondria “wasting” 50% of substrates (= food) not to produce ATP but to dissipate heat seems unlikely except in the case of a thermogenic tissue (e.g. brown adipose tissue of small mammals).

In my experience of working with avian red blood cells in various species, I have never (myself doing the labwork or closely supervising) obtained LEAK that high, i.e. mean = 56%, range = [33-79%]. Here are some published (and unfortunately a lot of unpublished) values as a comparison:

- King penguin (adults): 28% [19-38%] (Stier et al. 2017)
- Pied flycatcher (adults): 17% [13-21%] (Stier et al. 2019)
- Pied flycatcher (chicks + adults) : 25% [10-41%] (Marciau & Stier, unpublished)
- Japanese quail (chicks + adults): 15% [4-25%] (Stier et al. submitted to FASEB)
- Great tit (chicks): 23% [14-32%] (Cossin-Sevrin & Stier, submitted to J Exp Biol)
- Zebra finch (chicks + adults): 25% [16-38%] (Stier & Monaghan, unpublished)
- 25 bird species (adults): 17% [10-30%] (Stier et al. unpublished)

People I collaborated with (but was not on site to closely supervise data collection) have sometimes obtained somewhat higher LEAK values:

- Great tit (chicks) : 30% [6-52%] (Casagrande et al. 2020)
- Zebra finch (juveniles): 38% [28-54%] (Ton et al. 2021)
- Collared flycatcher (chicks + adults): 19% [13-26%] (Zhu & Qvarnström, unpublished)
- Eastern Yellow Robin (adults): 24% [10-53%] (Sunnucks & Stier, unpublished)

People using the technique without collaboration also tended to have high LEAK values:

- Blue tit (adults): autumn 38% [15-82%] / winter 54% [1-98%] (Nord et al. 2021)

- Coal tit (adults) : autumn 43% [21-65%] / winter 53% [29-82%] (Nord et al. 2021)
- Great tit (adults): autumn 17% [1-52%] / winter 36% [19-68%] (Nord et al. 2021)
- Zebra finch (adults): young ~47% [30-75%] / old ~35% [17-48%] (Dawson & Salmon 2020)

Consequently, I am wondering if such high LEAK values are really biological, or more stemming from issues in the laboratory protocol. In Divakaruni & Brand 2011 you cite, usual proton leak is considered ~20%, and leak of 50+% are considered remarkable exceptions or potential methodological artefacts. This could potentially affect the validity of the results obtained (to an unknown extent), and gives a wrong idea to the reader without expertise in mitochondrial biology of the potential magnitude of the proton leak phenomenon.

I acknowledge that it is the data you currently have and there is no easy way to tease apart biology from methodological issue here, but this limitation should at least be mentioned and acknowledged in the manuscript.

>> The reviewer raises an interesting point about proton leak and the issues surrounding the accurate measurement of this parameter. In our protocol, we used freshly prepared aliquots of oligomycin at an appropriate concentration, and so we cannot pinpoint a methodological issue that would result in non-biologically high LEAK rates. It is well described that oligomycin increases the proton leak rate by inhibiting phosphorylation, which subsequently increases $\Delta\psi_m$. As such, the calculated coupling efficiency is underestimated. This error is usually less than 10%, but can be as high as 29% (Affourtit and Brand 2009, doi.org/10.1016/S0076-6879(09)05023-X). Applying this underestimated correction to our data, the true proton leak rates are in the range of 23-56% (if considering a maximum underestimation of coupling efficiency of 29%).

As the reviewer notes, published LEAK values fall within a very broad range, from ~20% up to as high as 98% of respiration. While many mammalian cell lines exhibit ~20% LEAK respiration (Divakaruni and Brand 2011, doi.org/10.1152/physiol.00046.2010), some can be as high as 70% (Affourtit and Brand 2009). Similarly, proton leak can account for ~50% of respiratory capacity in rat skeletal muscle (Rolfe and Brand 1996, doi.org/10.1152/ajpcell.1996.271.4.C1380). Our values of 33-79% fall within this broad range, and as such should be considered biological. Indeed, with LEAK respiration contributing to ~25% of organismal metabolic rate it is not surprising that LEAK respiration would be as high as 50% in some tissues (Rolfe and Brand 1996).

In conclusion, we are confident that we employed a sound methodology and that high LEAK rates should not be viewed as non-biological artefacts. This is of particular importance in systems where LEAK driven respiration may be employed for heat generation, as is the case here in zebra finch nestlings. It is also possible that the specific context of our study (high temperatures and developing young) could have contributed to higher leak values. However, taking into consideration the points above, we now include our LEAK respiration values in the discussion (L680-681).

6) **Adaptive programming of mitochondrial function?**

I am not sure that you are really able to test the adaptive nature of the programming of mitochondrial function by prenatal sound (what is stated/implied line 104, 293, 313-315, 342-343 for instance) with this study. To this aim, you should be able to provide evidence that mitochondrial changes in response to prenatal heat-call (vs. control playback) are improving individual response to heat-stress/challenge. This would involve for instance blocking this mitochondrial response (e.g. by increasing proton leak with DNP?), measuring the response of control vs. heat-call playback birds (control vs. DNP) to heat challenge (e.g. body T°C change/body mass loss / water loss) and showing that the mitochondrial changes due to heat-call playback indeed drive the enhanced performance in response to heat challenge. This would be quite tricky (but elegant) to perform (and not sure DNP would be the right

choice), but would be the proper way to prove adaptive programming through mitochondrial changes in my opinion.

Your data suggests that mitochondrial changes (i.e. higher efficiency due to higher OXPHOS, not lower LEAK) might be adaptive at high temperature (i.e. less heat produced per unit of ATP produced), but this remains to be really tested in my opinion. Consequently, I would suggest to be more careful in your phrasing about adaptive programming all along the manuscript.

We now better explained throughout the manuscript how the changes we observed in mitochondrial function can bring adaptive benefits, and underlie the adaptive changes in growth previously reported in Mariette & Buchanan 2016 (e.g. L636-657).

We thank the reviewer for his suggestion of further experimental work, which is nonetheless well beyond the scope of this study.

Other comments:

Nomenclature: I was told previously that it should be *ROUTINE* and *LEAK* instead of ROUTINE and LEAK, and that other mito parameters (ETS, OXPHOS, FCR) are supposed to be italicized as well (e.g. to avoid confusion between OXPHOS = the process of oxidative phosphorylation vs. *OXPHOS* = rate of respiration under phosphorylating conditions).

>> We now italicised all mitochondrial parameters throughout the manuscript and supplementary materials as suggested.

Lines 26-27: Prenatal heat-call did not shifted away respiration from proton leak since LEAK respiration was not affected by your prenatal playback, while OXPHOS was stimulated.

Please reformulate

>> This sentence has been deleted.

Lines 27-28 : In my opinion, you cannot say that since there is no statistical support for a disruption of the impact of prenatal heat calls on mitochondrial function by post-natal heat challenge, i.e. no significant playback*heat-challenge interaction.

>> This sentence has been deleted.

Line 41: early-life instead of early?

>> Corrected (L57).

Line 50: mitochondrial function instead of mitochondria?

>> Corrected (L66).

Lines 48-50 + lines 62-63: the fact that it is unknown in animal is not fully correct since as you mentioned later on that Noguera et al. showed that prenatal acoustic cues could influence mitochondrial density (mtDNA copy number). This should be mentioned here as a first hint toward the prenatal programming of mitochondrial phenotype by acoustic cues.

>> Changes in mtDNA copy number could be due to many other changes than changes in mitochondrial functions. We left the text unchanged.

Lines 94-95: the fact that proton leak is an essential source of heat production in birds has never been fully proven to the best of my knowledge, maybe rephrase to be less assertive?

>> Corrected (L110-111)

Lines 119-121: there is no real direction in this prediction, so is it really useful?

>> All predictions were reformulated (L161-173)

Line 164: decapitated instead of decapitation?

>> Both words are correct, we were using the noun and not the verb. We added a comma for clarifying the grammar.

Lines 167 & 189 : you analysed 46 samples but present data for 41 individuals. The reason for excluding these 5 samples should be explained in my opinion.

>> We now provided this information (L286-289).

Lines 166-168 : what T°C have you used for measuring mitochondrial respiration?

>> The information was moved from the supplementary materials to the main text (L256).

Lines 212: maybe specify 10% of what (total N I guess)

>> Specified (L345-346).

Results : please be careful to always specify “was not significantly different” for all $p > 0.05$ (or 0.10 since you discuss ‘marginal’ effects). It is important since we cannot say there is no effect based on $p\text{-val} > 0.05$

>> We specified significance throughout the result section.

Lines 282-284 : this would even argue against the adaptive nature of the prenatal programming, no?

>> Sentence deleted.

Lines 285-286 : your more robust test for short-term effects of ambient temperature is the controlled heat-challenge you performed, and this had no significant effect on any of the mitochondrial parameters. This should be highlighted here in my opinion.

>> Sentence deleted.

Line 293 : as mentioned in major comment 6, I would rephrase to be more cautious here.

>> We have now clarified the adaptive values of effects we evidenced.

Line 297 : mitochondria do not generate energy. They convert energy from one source (food) to other forms of energy (ATP + heat).

>> Sentence deleted.

Line 310: neither your study nor the alligator one are wildlife studies since they were conducted on captive animals.

>> “Wildlife” refers to non-domesticated animal species, whether they are free-living or in captivity. In the alligator study (Galli et al 2016), eggs were directly collected from wild nests. In our study, zebra finches are wild-derived individuals (10th generation) and not domestic birds.

Lines 313-315: please rephrase according to major comment 6.

>> We have now clarified the adaptive values of effects we evidenced.

Lines 317-319: exposure to acute heat challenge had no significant effect, so you cannot state that in my opinion.

>> Sentence deleted.

Lines 319-321: exposure to acute heat challenge had no significant effect, so you cannot state that in my opinion.

>> Sentence deleted.

Lines 333-335: I don't think that you can state that [15] found a positive association between ATP production efficiency and growth performance. This paper reports the effects of oral CORT on both growth and mitochondrial function, but not on the relationship between growth and mitochondrial function.

>> Sentence deleted.

Lines 335-338: in my opinion it is maybe too speculative to imply that.

>> Sentence deleted.

Line 342: please rephrase according to major comment 6

>> Our conclusion is correct, given our better presentation of the adaptive values of the effects we demonstrated.

References (not cited in the manuscript):

Wolff, J. N. et al. *Evolutionary implications of mitochondrial genetic variation: mitochondrial genetic effects on OXPHOS respiration and mitochondrial quantity change with age and sex in fruit flies. Journal of Evolutionary Biology* 29, 736–747 (2016).